



# An Uncertainty Partition Approach for Inferring Interactive Hydrologic Risks

Yurui Fan[1], Kai Huang[2], Guohe Huang[3], Yongping Li[4], Feng Wang[4]

[1] Department of Civil and Environmental Engineering, Brunel University, London, Uxbridge, Middlesex, UB8 3PH, United Kingdom
[2] Faculty of Engineering and Applied Sciences, University of Regina, Regina, SK, Canada, S4S0A2
[3] Institute for Energy, Environment and Sustainable Communities, University of Regina, Regina, Saskatchewan, Canada S4S 0A2
[4] School of Environment, Beijing Normal University, Beijing 100875, China

*Correspondence to*: Yurui Fan (yurui.fan@brunel.ac.uk), Guohe Huang (huangg@uregina.ca)

## Abstract:

Extensive uncertainties exist in hydrologic risk analysis. Particularly for interdependent hydrometeorological extremes, the random features in individual variables and their dependence structures may lead to bias and uncertainty in future risk inferences. In this study, a full-subsampling factorial copula (FSFC) approach is proposed to quantify parameter uncertainties and further reveal their contributions to predictive uncertainties in risk inferences. Specifically, a full-subsampling factorial analysis (FSFA) approach is developed to diminish the effect of the sample size and provide reliable characterization for parameters' contributions to the resulting risk inferences. The proposed approach is applied to multivariate flood risk inference for Wei River basin to demonstrate the applicability of FSFC for tracking the major contributors to resulting uncertainty in a multivariate risk analysis framework. In detail, the multivariate risk model associated with flood peak and volume will be established and further introduced into the proposed full-subsampling factorial analysis framework to reveal the individual and interactive effects of parameter uncertainties on the predictive uncertainties in the resulting risk inferences. The results suggest that uncertainties in risk inferences would mainly be attributed to some parameters of the marginal distributions while the parameter of dependence structure



(i.e. copula function) would not produce noticeable effects. Moreover, compared with traditional

factorial analysis (FA), the proposed FSFA approach would produce more reliable visualization for

parameters' impacts on risk inferences, while the traditional FA would remarkable overestimate

contribution of parameters' interaction to the failure probability in AND, and at the same time,

underestimate the contribution of parameters' interaction to the failure probabilities in OR and Kendall.

# 1. Introduction

Many hydrological and climatological extremes are highly correlated among each other, and it is desired

to explore their interdependence through multivariate approaches. Examples include sea level rise and

fluvial flood (Moftakhari et al., 2017), drought and heat waves (Sun et al., 2019), soil moisture and

precipitation (AghaKouchak, 2015). Moreover, even one specific hydrological extreme may have

multiple attributes, such as the peak and volume for a flood, duration and severity for a drought, and

duration and intensity of a storm (Karmakar and Simonovic, 2009; Kong et al., 2019). Traditional

univariate approaches, mainly focusing on one variable or one attribute of hydrological extremes (e.g.

flood peak), may not be sufficient to describe those hydrological extemes containing multivariate

characteristics. Thus the univariate frequency/risk analysis methods may be unable to obtain reliable

risk inferences for the failure probability or recurrence intervals of interdependent extreme events

(Chebana and Ouarda, 2011; Requena et al., 2013; Salvadori et al., 2016; Sadegh et al., 2017)

Since the introduction of copula function into hydrology and geosciences by De Michele and Salvador

(2003), the copula-based approaches have been widely used for multivariate hydrologic risk analysis.

The copula functions are able to model correlated variables with complex or nonlinear dependence

structures. Also, this kind of methods are easily to be implemented since the marginal distributions and





dependence model can be estimated in separate processes, which also give flexibility in selection of

both marginal and dependence models. A great number of research has been developed for multivariate

hydrologic simulation through copula functions, such as multivariate flood frequency analysis (Sraj et

al., 2014; Xu et al., 2016; Fan et al., 2018); drought assessments (Song and Singh 2010; Kao and

Govindaraju 2010; Ma et al. 2013); storm or rainfall dependence analysis (Zhang and Singh 2007;

Vandenberghe et al. 2010); streamflow simulation (Lee and Salas 2011; Kong et al., 2015) and other

water and environmental engineering applications (Fan et al., 2017; Huang et al., 2017).

For both univariate and multivariate analysis for hydrometeorological risks, uncertainty would be one of

the unavoidable issues which needs to be well addressed. The uncertainty in hydrometeorological risk

inference mainly results from stochastic variability of hydrometeorological processes and incomplete

knowledge of the watershed systems (Merz and Thieken, 2005). Amounts of studies have been proposed

to address uncertainty in both univariate and multivariate hydrological risk analysis (e.g. Merz and

Thieken, 2005; Serinaldi, 2013; Dung et al., 2015; Zhang et al., 2015; Sadegh et al., 2017; Fan et al.,

2018). However, one critical issue in uncertainty quantification of hydrological inference is how to

characterize the major sources for uncertain risk inference. Qi et al. (2016) employed a subsampling

ANOVA approach (Bosshard et al., 2013), to quantify individual and interactive impacts of the

uncertainties in data, probability distribution functions, and probability distribution parameters on the

total cost for flood control in terms of flood peak flows. Even though the subsampling ANOVA

approach is able to reduce the effect of the biased estimator on quantification of variance contribution

resulting from traditional ANOVA approach, it would be noticed that merely subsampling one

uncertainty parameter/factor (referred as single-subsampling ANOVA), as used in the studies by

Bosshard et al. (2013) and Qi et al. (2016a), will lead to underestimation of individual contribution for

the factor to be sampled and overestimation of contributions for those non-sampled factors. Moreover,

few studies have been reported to characterize the individual and interactive effects of parameter




uncertainties in marginal and dependence models on the multivariate risk inferences.


Consequently, as an extension of previous research, this study aims to propose a Full-subsampling factorial copula (FSFC) approach for uncertainty quantification and partition in multivariate hydrologic risk inference. In detail, the parameter uncertainties are quantified through a Monte Carlo-based Bootstrap algorithm. The interactions of parameter uncertainties are explored through a multilevel

factorial analysis approach. The contributions of parameter uncertainties are analyzed through a full-subsampling factorial analysis (FSFA) method, in which all uncertain factors will be subsampled to generate more reliable results. The applicability of the proposed FSFC approach will demonstrated through case studies of flood risk analysis in the Wei River basin in China.

## 2. Methodology


Figure 1 illustrates the framework of the proposed FSFC approach. The framework consists of four modules: (i) selection of marginal distributions, (ii) identification of copulas, (iii) parameter uncertainty quantification, (iv) parameter interaction and sensitivity analysis. In FSFC, modules (i) and (ii) are proposed to construct the most appropriate copula-based hydrologic risk model. Module (iii) quantifies

parameter uncertainties in marginal distributions and copulas. Modules (iv) would be the core part of our study to identify the main sources of uncertainties in multivariate risk inference by the proposed full-subsampling factorial analysis (FSFA) approach.

----------------------------------------

Place Figure 1 here

----------------------------------------



## 2.2. Copula-based Multivariate Risk Inference Framework

A copula function is a multivariate distribution function with uniform margins on the interval [0, 1]. Sklar's Theorem states that any $d$-dimensional distribution function $F$ can be formulated through a copula and its marginal distributions (Nelsen, 2006). In detail, a multivariate copula function can be expressed as:

$$F(x_1, x_2, ..., x_d \mid \gamma_1, \gamma_2, ..., \gamma_d, \theta) = C(F_{X_1}(x_1 \mid \gamma_1), F_{X_2}(x_2 \mid \gamma_2), ..., F_{X_d}(x_d \mid \gamma_d) \mid \theta) \qquad (1)$$

where $F_{X_1}(x_1 \mid \gamma_1), F_{X_2}(x_2 \mid \gamma_2), ..., F_{X_d}(x_d \mid \gamma_d)$ are marginal distributions of the random vector $(X_1, X_2, …, X_d)$, with $\gamma_1, \gamma_2, …, \gamma_d$ respectively being the unknow parameters of marginal distributions. $\theta$ is the parameter in the copula function describing dependence among the correlated variables. If these marginal distributions are continuous, then a single copula function C exists, which can be written as (Nelsen, 2006):

$$C(u_1, u_2, ..., u_d \mid \theta) = F(F_{X_1}^{-1}(u_1 \mid \gamma_1), F_{X_2}^{-1}(u_2 \mid \gamma_2), ..., F_{X_d}^{-1}(u_d \mid \gamma_d)) \qquad (2)$$

where $u_1 = F_{X_1}(x_1 \mid \gamma_1)$, $u_2 = F_{X_2}(x_2 \mid \gamma_2)$, …, $u_d = F_{X_d}(x_d \mid \gamma_d)$. More details on the theoretical background and properties of various copula families can be found in Nelsen (2006).

If appropriate copula functions are specified to reflect the joint probabilistic characteristics among for a

multivariate extreme event, the conditional, primary and secondary return periods (RP) can be obtained. Consider one kind of hydrological extreme (denoted as $X$) with $d$ attributes (i.e. $X = (x_1, x_2, …, x_d)$), and for a specific extreme event $X^*$ with its attributes being $X^* = (x_1^*, x_2^*, …, x_d^*)$, three categories of multivariate RP can be applied for revealing the potential risk of $X^*$.

(i) "OR" case:



$$T^{OR} = \{(x_1, x_2, ..., x_d) \in R^d : x_1 > x_1^* \vee x_2 > x_2^* \vee ... \vee x_d > x_d^*\}$$

$$= \frac{\mu}{1 - C(F_1(x_1 \mid \gamma_1), ..., F_d(x_d \mid \gamma_d) \mid \theta)} \tag{4}$$

where $\mu$ denotes the average time between two adjacent events under consideration.

(ii) "AND" case:

$$T^{AND} = \{(x_1, x_2, ..., x_d) \in R^d : x_1 > x_1^* \wedge x_2 > x_2^* \wedge ... \wedge x_d > x_d^*\}$$

$$= \frac{\mu}{\hat{C}(\overline{F}_1(x_1 \mid \gamma_1), \overline{F}_2(x_2 \mid \gamma_2), ..., \overline{F}_1(x_d \mid \gamma_d) \mid \theta)} \tag{5}$$

where $\hat{C}$ is the multivariate survival function of the $X_i$'s proposed by Salvadori et al. (2013; 2016),

and $\overline{F}_i(x_i \mid \gamma_i) = P(X > x_i) = 1 - F_i(x_i \mid \gamma_i)$. Following Salvadori et al. (2013; 2016), and the Inclusion-

Exclusion principle proposed by Joe (2014), the multivariate survival function $\hat{C}$ can be obtained by:

$$\hat{C}(\mathbf{u}) = \overline{C}(1 - \mathbf{u}) \tag{6}$$

and

$$\overline{C}(\mathbf{u}) = 1 - \sum_{i=1}^d u_i + \sum_{S \in P} (-1)^{\#(S)} C_S(u_i : i \in S) \tag{7}$$

(iii) "Kendall" case: The Kendall RP characterizes the hydrologic disasters exceeding a critical layer as

defined by (Salvadori et al., 2011): $L_t^F = \{x \in R^d : F(x) = t\}$. The Kendall RP can be expressed as

(Salvadori et al., 2011):

$$T^{Kendall} = \frac{\mu}{1 - K_C(t)} \tag{8}$$

where $K_C$ is the Kendall distribution function associated with C, which can be expressed as:




$$K_C(t) = P(C(F_1(x_1 \mid \gamma_1), ..., F_d(x_d \mid \gamma_d) \mid \theta) \leq t) \tag{9}$$

In addition to the multivariate RP, Failure probability (FP) can be another index to provide more coherent, general and well devised tools for multivariate risk assessment and communication. In general, the failure probability $p_M$ to indicate the occurrence of a critical event for at least one time in $M$ years of design life can be defined as (Salvadori et al., 2016):

$$p_M = 1 - \prod_{j=1}^{M}(1 - p_j) = 1 - (F(x_d))^M \tag{10}$$

Similar to the multivariate RP concept, the failure probability in a multivariate context can also be characterized in "OR", "AND", and "Kendall" scenarios expressed by the following equations. For a given critical threshold $x^* = \{x_1^*, x_2^*, ..., x_d^*\}$, the failure probabilities violating this critical value can be expressed as (Salvadori et al., 2016):

$$p_T^{OR} = 1 - (C(F_1(x_1^* \mid \gamma_1), F_1(x_2^* \mid \gamma_2), ..., F_d(x_d^* \mid \gamma_d) \mid \theta))^T \tag{11a}$$

$$p_T^{AND} = 1 - (1 - \hat{C}(\overline{F}_1(x_1^* \mid \gamma_1), \overline{F}_2(x_2^* \mid \gamma_2), ..., \overline{F}_1(x_d^* \mid \gamma_d) \mid \theta))^T \tag{11b}$$

$$p_T^{Kendall} = 1 - (P(C(F_1(x_1^* \mid \gamma_1), F_1(x_2^* \mid \gamma_2), ..., F_d(x_d^* \mid \gamma_d) \mid \theta) \leq t))^T \tag{11c}$$

where $p_T^{OR}$, $p_T^{AND}$, and $p_T^{Kendall}$ respectively denote the failure probability in "AND", "OR" and "Kendall" cases. $T$ indicates the service time of the facilities under consideration.

Focusing on a bivariate case, the joint RP and the associate failure probability in "OR", "AND", and "Kendall" scenarios can be formulated as (Salvadori et al., 2007, 2011; Graler et al., 2013; Sraj et al., 2014; Serinaldi, 2015):

$$T_{u_1,u_2}^{OR} = \frac{\mu}{1 - C_{U_1 U_2}(u_1, u_2 \mid \theta)} \tag{12a}$$





$$T_{u_1,u_2}^{AND} = \frac{\mu}{1 - u_1 - u_2 + C_{U_1 U_2}(u_1, u_2 \mid \theta)} \tag{12b}$$

$$T_{u_1,u_2}^{Kendall} = \frac{\mu}{1 - P(C_{U_1 U_2}(u_1^*, u_2^*) \leq t)} \tag{12c}$$

$$p_T^{OR} = 1 - (C_{U_1 U_2}(u_1^*, u_2^* \mid \theta))^T \tag{12d}$$

$$p_T^{AND} = 1 - (u_1^* + u_2^* - \hat{C}_{U_1 U_2}(u_1^*, u_2^* \mid \theta))^T \tag{12e}$$

$$p_T^{Kendall} = 1 - (P(C_{U_1 U_2}(u_1^*, u_2^* \mid \theta) \leq t))^T \tag{12f}$$

where $u_1 = F_1(x_1 \mid \gamma_1), u_2 = F_2(x_2 \mid \gamma_2)$, $u_1^* = F_1(x_1^* \mid \gamma_1), u_2^* = F_2(x_2^* \mid \gamma_2)$, $(x_1^*, x_2^*)$ defines the bivariate

threshold.

## 2.3. Uncertainty in the Copula-base Risk Model

Extensive uncertainties may be involved in the parametric estimation of a copula function due to: (i) the

inherent uncertainty in the flooding process; (ii) uncertainty in the selection of appropriate marginal

functions and copulas; and, (iii) statistical uncertainty or parameter uncertainty within the parameter

estimation process (e.g. the availability of samples) (Zhang et al., 2015). Several methods have been

proposed to quantify parameter uncertainties in copula-based models. For instance, Dung et al. (2015)

proposed bootstrap-based methods for quantifying the parameter uncertainties in bivariate copula

models. Zhang et al. (2015) employed a Bayesian inference approach for evaluating uncertainties in

copula-based hydrologic droughts models, in which the Component wise Hit-And-Run Metropolis

algorithm is adopted to estimate the posterior probabilities of model parameters.





In this study, a bootstrap-based algorithm, will be applied to quantify parameter uncertainties in the

copula-based multivariate risk model. The procedures the bootstrap-based algorithm to derive

probabilistic distributions of the parameters in both marginal and dependence models are presented as

follows:

1. Predefine a large number of bootstrapping samplings $N_B$

2. Implement the resampling with replacement over observed pairs $Z = (X, Y)$ to obtain $Z^* = (X^*, Y^*)$.

$Z^*$ has the same size as $Z$

3. Fit the chosen marginal distributions to $X^*$ and $Y^*$, and estimate the associated parameters ($\gamma_X, \gamma_Y$).

4. Fit the chosen copula to $Z^*$, and estimate the parameter in the copula function $\theta$.

5. Repeat step 2–5 $N_B$ times, and obtain $N_B$ sets of $(\gamma_X, \gamma_Y, \theta)$. Moreover, the reject those parameters

that lead to bad fits for both marginal and copula models, the A-D test and the Cramer-von-Mises test

are introduced in the bootstrap procedure to ensure that the obtained parameters can pass statistic tests

for both the marginal distribution and copula models. Then the kernel method will be adopted to

quantify the probabilistic features for $\gamma_X, \gamma_Y, \theta$.

6. In order to derive bivariate uncertainty bands for a predefined quantile curve ($QC$) with certain joint

RP in 'AND', 'OR' or 'Kendall' (denoted as $T^{AND}$, $T^{OR}$, $T^{Kendall}$), sample $N_{B_1}$ sets of $(\gamma_X, \gamma_Y, \theta)$ from

the obtained $N_B$ samples

7. Sample a large number ($N_s$) of $x_i$ $y_j$ from their marginal distributions.

8. For each set of $(\gamma_X, \gamma_Y, \theta)$ from $N_{B_1}$, evaluate the joint RPs of $(x_i, y_j)$ ($i = 1, 2, …, N_s; y = 1, 2, …,$

$N_s$), and store the pairs of $(x_i, y_j)$ approaching the predefined joint RPs.

9. Repeat step 8 for $N_{B_1}$, and for each predefined $QC$, and plot the bivariate uncertainty bands for each

quantile $QC$



## 2.4. Interactive and Sensitivity Analysis for Parameter Uncertainties

Due to the uncertainties existing in the unknown parameters for a copula model, the associated risk or

the return period for a flooding event may also be uncertain. Few studies have been reported to analyze

the effect of uncertainties in the copula model on evaluating the risk for a flood event. To address the

above issue, a full-subsampling factorial analysis (FSFA) approach will be proposed to reveal the

individual and interactive effects of parameter uncertainties on the predictive uncertainties of different

risk inferences.

Consider a copula-based bivariate risk assessment model which has two marginal distributions ($A$ and

$B$) and one copula ($C$). The parameters in the two marginal distributions are assumed to be respectively

denoted as $\gamma^A$ with $a$ levels and $\gamma^B$ with b levels, while the parameter in the copula is denoted with $\theta^C$

with $c$ levels. The three factor ANOVA model for such a factorial design in terms of the predictive risk

(denoted as $R$) in response to the parameters $\gamma_A$, $\gamma_B$, $\theta_C$ and $n$ replicates, can be expressed as:

$$R_{ijkl} = R_0 + R_{\theta_i^C} + R_{\gamma_j^A} + R_{\gamma_k^B} + R_{\theta_i^C \gamma_j^A} + R_{\theta_i^C \gamma_k^B} + R_{\gamma_j^A \gamma_k^B} + R_{\theta_i^C \gamma_j^A \gamma_k^B} + \varepsilon_{ijkl} \begin{cases} i = 1, 2, ..., c \\ j = 1, 2, ..., a \\ k = 1, 2, ...b \\ l = 1, 2, ..., n \end{cases} \quad (13)$$

where $R_0$ denotes the overall mean effect; $R_{\theta_i^C}, R_{\gamma_j^A}, R_{\gamma_k^B}$ respectively indicate the effect for parameter $\theta^C$

in the copula at the $i$th level, parameter $\gamma^A$ in the first marginal distribution at the $j$th level, and parameter

$\gamma^B$ in the first marginal distribution at the $k$th level; $R_{\theta_i^C \gamma_j^A}, R_{\theta_i^C \gamma_k^B}, R_{\gamma_j^A \gamma_k^B}$ indicate interactions between

factors $\theta^C$ and $\gamma^A$, $\theta^C$ and $\gamma^B$, as well as $\gamma^A$ and $\gamma^B$, respectively; $R_{\theta_i^C \gamma_j^A \gamma_k^B}$ denotes the interaction of factors

$\theta^C$, $\gamma^A$ and $\gamma^B$; $\varepsilon_{ijkl}$ denotes the random error component.



Based on Equation (13), the total variability of the predictive risk can be decomposed into its
component parts as follows (Montgomery, 2001):

$$SS_T = SS_{\theta^C} + SS_{\gamma^A} + SS_{\gamma^B} + SS_I + SS_e \tag{14a}$$

and

$$SS_T = \sum_{i=1}^{c}\sum_{j=1}^{a}\sum_{k=1}^{b}\sum_{l=1}^{n} R_{ijkl}^2 - \frac{R_{....}^2}{abcn} \tag{14b}$$

$$SS_{\theta^C} = \frac{1}{abn}\sum_{i=1}^{c} R_{i...}^2 - \frac{R_{....}^2}{abcn} \tag{14c}$$

$$SS_{\gamma^A} = \frac{1}{bcn}\sum_{j=1}^{a} R_{.j..}^2 - \frac{R_{....}^2}{abcn} \tag{14d}$$

$$SS_{\gamma^B} = \frac{1}{acn}\sum_{k=1}^{b} R_{..k.}^2 - \frac{R_{....}^2}{abcn} \tag{14e}$$

$$SS_e = \sum_{i=1}^{c}\sum_{j=1}^{a}\sum_{k=1}^{b}\sum_{l=1}^{n} R_{ijkl}^2 - \frac{1}{n}\sum_{i=1}^{c}\sum_{j=1}^{a}\sum_{k=1}^{b} R_{ijk.}^2 \tag{14f}$$

$$\begin{aligned} SS_I &= SS_{\theta^C\gamma^A} + SS_{\theta^C\gamma^B} + SS_{\gamma^B\gamma^A} + SS_{\theta^C\gamma^A\gamma^B} \\ &= SS_T - SS_{\theta^C} - SS_{\gamma^A} - SS_{\gamma^B} - SS_e \end{aligned} \tag{14g}$$

where $R_{ijk.} = \sum_{l=1}^{n} R_{ijkl}$, $R_{i...} = \sum_{j=1}^{a}\sum_{k=1}^{b}\sum_{l=1}^{n} R_{ijkl}$, $R_{.j..} = \sum_{i=1}^{c}\sum_{k=1}^{b}\sum_{l=1}^{n} R_{ijkl}$, $R_{..k.} = \sum_{i=1}^{c}\sum_{j=1}^{a}\sum_{l=1}^{n} R_{ijkl}$

$R_{....} = \sum_{i=1}^{c}\sum_{j=1}^{a}\sum_{k=1}^{b}\sum_{l=1}^{n} R_{ijkl}$ . Then the contributions of parameter uncertainties in marginal

distributions and dependence structures can be calculated as:

(1) Contribution of parameters in marginal distributions A and B





$$\eta_A = SS_{\gamma^A} / SS_T \tag{15a}$$

$$\eta_B = SS_{\gamma^B} / SS_T \tag{15b}$$

(2) Contribution of the parameter in the dependence structure

$$\eta_C = SS_{\theta^C} / SS_T \tag{15c}$$

(3) Contribution of internal variability

$$\eta_e = SS_e / SS_T \tag{15d}$$

(4) Contribution of parameter interactions

$$\eta_I = 1 - \eta_A - \eta_B - \eta_C - \eta_e \tag{15e}$$

However, one major issue for ANOVA approach is that the biased variance estimator in ANOVA would
underestimate the variance in small sample size scenarios (Bosshard et al., 2013). Thus the sample size
may significantly affect the resulting variance contributions expressed in Equations (15a) – (15e). A
subsampling approach has been advanced by Bosshard et al. (2013) to diminish the effect of the sample
size in ANOVA and has been employed for uncertainty partition in flood design and hydrological
simulation (Qi et al., 2016a, b). In such a subsampling scheme, one factor (denoted as X) with T levels
(these levels can be different values for numerical parameters, or different types for non-numerical
factor (e.g. model type)), would choose two levels in each iteration. For $T$ possible levels of X, we can
obtain a total of $C_T^2$ possible pairs for X, expressed as a $2 \times C_T^2$ matrix as follows:

$$g(h, j) = \begin{pmatrix} X_1 & X_1 & \cdots & X_1 & X_2 & X_2 & \cdots & X_{T-2} & X_{T-2} & X_{T-1} \\ X_2 & X_3 & \cdots & X_T & X_3 & X_4 & \cdots & X_{T-1} & X_T & X_T \end{pmatrix} \tag{16}$$

However, such a subsampling approach mainly applied to subsample merely one factor or one





parameter (here we refer to this method as single-subsampling ANOVA) in previous studies (Bosshard et al. 2013; Qi et al., 2016a, b). However, one critical issue for the single-subsampling ANOVA it that it

will lead to underestimation of individual contribution for the factor to be sampled and overestimation of contributions for those non-sampled factors. Consequently, in this study, we will propose a FSFA approach to subsample all the factors to be addressed, and then quantify the contribution of each factor to the response variation. In the FSFA approach, all factors under consideration will be subsampled, and the corresponding sum of squares will be obtained. The contribution of one factor would be

characterized by the mean value of its contribution in each iteration. In detail, for the three factor ANOVA model expressed by Equation (13), the subsampling schemes for the three parameters can be formulated as:

$$g_{\theta^C}(h_C, j_C) = \begin{pmatrix} \theta_1^C & \theta_1^C & \cdots & \theta_1^C & \theta_2^C & \theta_2^C & \cdots & \theta_{c-2}^C & \theta_{c-2}^C & \theta_{c-1}^C \\ \theta_2^C & \theta_3^C & \cdots & \theta_c^C & \theta_3^C & \theta_4^C & \cdots & \theta_{c-1}^C & \theta_c^C & \theta_c^C \end{pmatrix} \quad (17a)$$

$$g_{\gamma^A}(h_A, j_A) = \begin{pmatrix} \gamma_1^A & \gamma_1^A & \cdots & \gamma_1^A & \gamma_2^A & \gamma_2^A & \cdots & \gamma_{a-2}^A & \gamma_{a-2}^A & \gamma_{a-1}^A \\ \gamma_2^A & \gamma_3^A & \cdots & \gamma_a^A & \gamma_3^A & \gamma_4^A & \cdots & \gamma_{a-1}^A & \gamma_a^A & \gamma_a^A \end{pmatrix} \quad (17b)$$

$$g_{\gamma^B}(h_B, j_B) = \begin{pmatrix} \gamma_1^B & \gamma_1^B & \cdots & \gamma_1^B & \gamma_2^B & \gamma_2^B & \cdots & \gamma_{b-2}^B & \gamma_{b-2}^B & \gamma_{b-1}^B \\ \gamma_2^B & \gamma_3^B & \cdots & \gamma_b^B & \gamma_3^B & \gamma_4^B & \cdots & \gamma_{b-1}^B & \gamma_b^B & \gamma_b^B \end{pmatrix} \quad (17c)$$

Consequently, there are a total number of $C_c^2 C_a^2 C_b^2$ iterations in FSFA for the three-factor model expressed as Equation (13). For each iteration, the sums of squares can be reformulated as:

$$SS_T^j = \sum_{h_C=1}^{2} \sum_{h_A=1}^{2} \sum_{h_B=1}^{2} \sum_{l=1}^{n} R^2_{g_{\theta^C}(h_C,j_C) g_{\gamma^A}(h_A,j_A) g_{\gamma^B}(h_B,j_B) l} - \frac{R^2_{g_{\theta^C}(o,j_C) g_{\gamma^A}(o,j_A) g_{\gamma^B}(o,j_B).}}{8n} \quad (18a)$$

$$SS_{\theta^C}^j = \frac{1}{4n} \sum_{h_C=1}^{2} R^2_{g_{\theta^C}(h_C,j_C) g_{\gamma^A}(o,j_A) g_{\gamma^B}(o,j_B).} - \frac{R^2_{g_{\theta^C}(o,j_C) g_{\gamma^A}(o,j_A) g_{\gamma^B}(o,j_B).}}{8n} \quad (18b)$$





$$SS_{\gamma^A}^j = \frac{1}{4n} \sum_{h_A=1}^{2} R_{g_{\theta^C}(h_C,o)g_{\gamma^A}(h_A,j_A)g_{\gamma^B}(o,j_B).}^2 - \frac{R_{g_{\theta^C}(o,j_C)g_{\gamma^A}(o,j_A)g_{\gamma^B}(o,j_B).}^2}{8n} \tag{18c}$$

$$SS_{\gamma^B}^j = \frac{1}{4n} \sum_{h_B=1}^{2} R_{g_{\theta^C}(h_C,o)g_{\gamma^A}(o,j_A)g_{\gamma^B}(h_B,j_B).}^2 - \frac{R_{g_{\theta^C}(o,j_C)g_{\gamma^A}(o,j_A)g_{\gamma^B}(o,j_B).}^2}{8n} \tag{18d}$$

$$SS_e^j = \sum_{h_C=1}^{2}\sum_{h_A=1}^{2}\sum_{h_B=1}^{2}\sum_{l=1}^{n} R_{g_{\theta^C}(h_C,j_C)g_{\gamma^A}(h_A,j_A)g_{\gamma^B}(h_B,j_B)l}^2 - \frac{1}{n}\sum_{h_C=1}^{2}\sum_{h_A=1}^{2}\sum_{h_B=1}^{2} R_{g_{\theta^C}(h_C,j_C)g_{\gamma^A}(h_A,j_A)g_{\gamma^B}(h_B,j_B).}^2 \tag{18e}$$

$$SS_I^j = SS_T^j - SS_{\theta^C}^j - SS_{\gamma^A}^j - SS_{\gamma^B}^j - SS_e^j \tag{18f}$$

where

$$R_{g_{\theta^C}(h_C,j_C)g_{\gamma^A}(h_A,j_A)g_{\gamma^B}(h_B,j_B).} = \sum_{l=1}^{n} R_{g_{\theta^C}(h_C,j_C)g_{\gamma^A}(h_A,j_A)g_{\gamma^B}(h_B,j_B)l}$$

$$R_{g_{\theta^C}(h_C,j_C)g_{\gamma^A}(o,j_A)g_{\gamma^B}(o,j_B).} = \sum_{h_A=1}^{2}\sum_{h_B=1}^{2}\sum_{l=1}^{n} R_{g_{\theta^C}(h_C,j_C)g_{\gamma^A}(h_A,j_A)g_{\gamma^B}(h_B,j_B)l} \cdot$$

$$R_{g_{\theta^C}(o,j_C)g_{\gamma^A}(h_A,j_A)g_{\gamma^B}(o,j_B).} = \sum_{h_C=1}^{2}\sum_{h_B=1}^{2}\sum_{l=1}^{n} R_{g_{\theta^C}(h_C,j_C)g_{\gamma^A}(h_A,j_A)g_{\gamma^B}(h_B,j_B)l}$$

$$R_{g_{\theta^C}(o,j_C)g_{\gamma^A}(o,j_A)g_{\gamma^B}(h_B,j_B).} = \sum_{h_C=1}^{2}\sum_{h_A=1}^{2}\sum_{l=1}^{n} R_{g_{\theta^C}(h_C,j_C)g_{\gamma^A}(h_A,j_A)g_{\gamma^B}(h_B,j_B)l}$$

$$R_{g_{\theta^C}(o,j_C)g_{\gamma^A}(o,j_A)g_{\gamma^B}(o,j_B).} = \sum_{h_C=1}^{2}\sum_{h_A=1}^{2}\sum_{h_B=1}^{2}\sum_{l=1}^{n} R_{g_{\theta^C}(h_C,j_C)g_{\gamma^A}(h_A,j_A)g_{\gamma^B}(h_B,j_B)l}$$

Also, for each iteration, the corresponding contributions for each factor can be obtained as:

$$\eta_A^j = SS_{\gamma^A}^j / SS_T^j \tag{19a}$$

$$\eta_B^j = SS_{\gamma^B}^j / SS_T^j \tag{19b}$$

$$\eta_C^j = SS_{\theta^C}^j / SS_T^j \tag{19c}$$

$$\eta_e^j = SS_e^j / SS_T^j \tag{19d}$$

$$\eta_I^j = 1 - \eta_A^j - \eta_B^j - \eta_C^j - \eta_e^j \tag{19e}$$



Finally, the individual and interactive contributions for those factors can be obtained by averaging the

corresponding contributions in all iterations, expressed as:

$$\eta_A = \frac{1}{J}\sum_{j=1}^{J} SS_{\gamma^A}^{j} \Big/ SS_T^{j} \tag{20a}$$

$$\eta_B = \frac{1}{J}\sum_{j=1}^{J} SS_{\gamma^B}^{j} \Big/ SS_T^{j} \tag{20b}$$

$$\eta_C = \frac{1}{J}\sum_{j=1}^{J} SS_{\theta^C}^{j} \Big/ SS_T^{j} \tag{20c}$$

$$\eta_e = \frac{1}{J}\sum_{j=1}^{J} SS_e^{j} \Big/ SS_T^{j} \tag{20d}$$

$$\eta_I = \frac{1}{J}\sum_{j=1}^{J} \eta_I^{j} \tag{20e}$$

where $\quad J = C_c^2 C_a^2 C_b^2$

## 3. Applications

The proposed FSFC approach can be applied for various multivariate risk inference problems. In this

study, we will apply FSFC for multivariate flood risk inference at the Wei River basin in China. The

Weihe River plays a key role in the economic development of western China, and thus is known

regionally as the 'Mother River' of the Guanzhong Plain of the southern part of the loess plateau (Song

et al. 2007; Zuo et al. 2014; Du et al. 2015, Xu et al., 2016). It originates from the Niaoshu Mountain at

an elevation of 3485 m above mean sea level in Weiyuan County of Gansu Province (Du et al. 2015).

The Weihe River basin is characterized by a semi-arid and sub-humid continental monsoon climate,

resulting in significant temporal-spatial variations in precipitation, with an annual precipitation of 559

mm (Xu et al., 2016). Furthermore, there is a strong decreasing gradient from south to north, in which



the southern region experiences a sub-humid climate with annual precipitation ranging from 800 to
1000 mm, whereas the northern region has a semi-arid climate with annual precipitation ranging from
400 to 700 mm (Xu et al., 2016). Over the entire basin, the mean temperature ranges from 6 to 14 $^0$C,
the annual potential evapotranspiration fluctuates from 660 to 1,600, and the annual actual
evapotranspiration is about 500 mm (Du et al. 2015).

Observed daily streamflow data at Xianyang and Zhangjiashan gauging stations are applied for
hydrologic risk analysis. Figure 1 show the locations of these three gauging stations. Based on the daily
stream flow data, the flood peak applied is defined as the maximum daily flow over a period and the
associated flood volume is considered as the cumulative flow during the flood period. In this study, the
flood characteristics are obtained based on an annual scale. This means that one flood event is identified
in each year. The detailed method to identify the flood peak and the associated flood volume can be
found in Yue (2000, 2001). Table 1 shows some descriptive statistical values for the considered
variables (peak discharge, Q; hydrograph volume, V), in which 47 and 55 flood events are characterized
at the Xianyang and Zhangjiashan station, respectively.

340     -------------------------------
Place Figure 2 and Table 1 here
-------------------------------

## 4. Results Analysis

### 4.1. Model Evaluation and Selection

There are a number of potential probabilistic models for modelling individual flood variables and their
dependence structures. In this study, five alternative distributions, including gamma, generalized
extreme value (GEV), lognormal (LN), Pearson type III (P III), and log-Pearson type III (LP III)



distributions, are employed to describe the probabilistic features of the chosen flood variables (i.e. peak

and volume). Moreover, goodness-of-fit tests are performed through the indices of Kolmogorov-

Smirnov test (K-S test), root mean square error (RMSE) and Akaike Information Criterion (AIC), to

screen the performance of those potential models. The results are presented in Table 2. The results

indicate that all five parametric distributions can produce satisfactory results, with all p-values larger

than 0.05. However, it can be concluded that the GEV and lognormal approaches show best

performance for respectively modelling flood peak and volume at both gauging stations.

-------------------------------------------

Place Tables 2 here

360     -------------------------------------------

In addition, a total number of six copulas, including Gaussian, Student t, Clayton, Gumbel, Frank and

Joe copulas, are considered as the candidate models for quantifying the dependence structures for flood

peak-volume at Xianyang and Zhangjiashan gauging stations. Also, the goodness-of-fit statistics is

performed based on the Cramér von Mises statistic proposed by Genest et al. (2009). And the indices of

RMSE and AIC would be further employed to evaluate the performance of the obtained copulas and

identify the most appropriate ones. Table 3 shows statistical test results for the selected copulas. The

results show that, for the Zhangjiashan station, all candidate copulas except the Joe copula performed

well, while all six copulas would be able to provide satisfactory risk inferences at the Xianyang station.

Moreover, based on the values of RMSE and AIC, the Gumbel copula was chosen to model the

dependence of flood peak and volume at Zhangjiashan station, while the Joe copulas performed best at

the Xianyang station. Consequently, the Gumbel and Joe copula would be chosen in this study to further

characterize the uncertainty in model parameters and the resulting risks at Zhangjiashan and Xianyang

station, respectively.

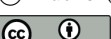




------------------------------

Place Tables 3 here

------------------------------


## 4.2. Uncertainty in Model Parameters and Risk Inferences

Based on the results in Tables 2 and 3, the multivariate risk inference model would be established, in

which the GEV and lognormal distributions would respectively be adopted to model the individual

flood variables at both gauging stations, while in comparison, the Gumbel and Joe copulas would

respectively be employed for Zhangjiashan and Xianyang stations. Afterward, uncertainties would be

characterized based on the bootstrap algorithm illustrated in Section 2.3. In current study, a total number

of 5000 samples would be chosen in order to generally visualize the uncertainty features in model

parameters. The probabilistic features for obtained parameters values (i.e. shape, scale and location for

GEV, meanlog, sdlog for LN, and theta for copula) for each sample scenario would be described by the

kernel method. Figure 3 exhibit the probabilistic distributions for the six unknow parameters in the

established multivariate risk inference model. Extensive uncertainties exist in the parameters for both

the marginal distribution and dependence model. As presented in Figure 3, each parameter, except the

meanlog in the LN distribution, exhibit noticeable uncertainty. Moreover, most of the parameter

uncertainties are approximately normally distributed.

------------------------------

Place Figure 3 here

------------------------------




It is quite apparent that different parameter values in the copula model would lead to different risk inference results. Consequently, parameter uncertainties in the marginal distributions and copula function would definitely result into uncertainties in multivariate risk inferences. Based on the copula model, some multivariate risk indices can be easily obtained, such as the joint return period in OR,

AND and Kendall, as expressed in Equations (12a) – (12c). However, due to parameter uncertainties, these risk indices may also exhibit some degrees of uncertainty. Figures 4 – 6 describe uncertainties for the joint RP in AND, OR and Kendall at the two stations. In general, the predictive RP in AND exhibit most significant uncertainty, followed by the predictive RP in OR and Kendall. However, for moderate or large flood events, considerable uncertainties can be observed in the inferences for all the three joint

RPs. Specifically, noticeable uncertainties exist in the predictive joint RP of AND even for a minor flood event with a 5-year joint RP. For some large flood events with a joint RP around 100 years, the predictive RP in AND shows remarkable uncertainty, ranging from less than 50 years to larger than 200 years. For the joint RP in OR and Kendall, slight uncertainty may exist for small flood events (e.g. 2-year or 5-year joint RP). Nevertheless, apparently uncertainties can be observed in the predictive joint

RP even for moderate flood events. As shown in Figure 5, considerable uncertainties may appear in the predictive joint RP of OR even for a flood with an actual joint RP of 20 years, while prediction of the Kendall RP for a 20-year (in Kendall RP) flood event may range from 10 to 50 years, as presented in Figure 6.

420 ---------------------------------
Place Figures 4-6 here
---------------------------------

## 4.3 Individual and Interactive Effects of Parameter Uncertainties


It has been observed that parameter uncertainties in the copula-based multivariate risk model would





lead to significantly imprecise risk predictions. However, one critical issue to be addressed is that how the parameter uncertainties and their interaction would influence the risk inference. Consequently, a multilevel factorial analysis, based on Equations (13) and (14) would be proposed to primarily visualize

the individual and interactive effects of parameter uncertainties in the marginal and dependence models on the resulting risk inferences. In this study, a total number of 6 parameters (i.e. three from GEV, two from LN, and one from copula) would to be addresses, and based on probabilistic features of these parameters, three quantile levels (i.e. 0.1, 0.5 and 0.9) would be chosen to characterize the resulting risk inferences under different parameter values. This would finally form a $3^6$ factorial design, which has six

factors with each having three levels. The failure probability denoted as Equations (11) would be considered as the responses in this factorial design.

The main and interactive effects of parameters uncertainties on the failure probabilities in AND are visualized in Figure 7. It is noticeable that at the two gauge stations, parameters uncertainties pose

similar main and interactive effects on the failure probabilities in AND, which indicates that parameters' effects (individual and interactive) on the failure probability in AND are independent with the location of gauge stations. More specifically, variations in the shape parameter in GEV and sdlog parameter in LN would lead to more changes in the corresponding responses (i.e. failure probability in AND) than the variations in other parameters. Also, as shown in Figure 7, the parameter in the copula function (i.e.

Cop_theta), describing dependence of the two flood variables, would not have an effect on the resulting risk as visible as the effects from the parameters (except the location parameter in GEV) in the marginal distributions. In terms of parameter interactions, the significance of interactive effects for different parameters is various. The interactive curves for some parameters (e.g. GEV_shape and GEV_location) are nearly parallel at the three levels, indicating an insignificant interaction for these two parameters on

the inferred risk. In comparison, there are also some interactive curves intersecting among each other (e.g. GEV_shape and LN_meanlog), implying a significant interaction among these two parameters.





Table 4 provides the results from an ANOVA table for the failure probability in AND. It is quite interesting that: i) even though the effect from the parameter in the copula function is not as visible as the effects from the parameters (except the location parameter in GEV) in the marginal distributions (as shown in Figure 7), such an effect is still statistically significant; i) the effect from the location parameter of GEV is statistically insignificant, which also lead to insignificant interactive effects between the location parameter and other parameters; iii) the interactions between the parameter in copula and the parameters in marginal distributions would be more likely statistically insignificant; iv) the statistical significance (significant or not) for individual and interactive effects from parameters is almost the same between these two gauge stations. All these conclusions obtained from Table 4 are consistent with the implications described in Figure 7.

--------------------------------

Place Figures 7 and Table 4 here

--------------------------------

In terms of the failure probabilities in OR and Kendall, as presented in Figures 8 and 9, these have similar pattern with the failure probability in AND (presented in Figure 7). The individual/main effects from the marginal distributions (except the location parameter in GEV) are generally more visible than the parameter in copula. Also, some interactive curves, especially the curves between GEV_location and others, are parallel, showing insignificant interaction between those parameters. More detailed characterization of the main and interactive effects for the failure probabilities in OR and Kendall is described in the ANOVA tables in Tables 5 and 6. These two tables show some slight differences from the conclusions given by Table 4. The location parameter in GEV also pose statistically significant effect on the results failure probabilities in OR and Kendall, which also leads to some significant interactions between this parameter and other model parameters. For the failure probability in Kendall,



the parameter in copula would have more interactions with other parameters in marginal distributions than the interactions in the failure probability in AND and OR. As presented in Table 6, the parameter in copula would have statistically significant effect on the inferred failure probability in Kendall with other

parameters except the location parameter in GEV. These results are also implied in the main effects plots and full interactions plot matrices in Figures 8 and 9.

--------------------------------------------
Place Figures 8-9 and Table 5 - 6 here

485   --------------------------------------------

Based on the three-level factorial analysis, it can be generally concluded that the parameters in marginal distributions (except the location parameter in GEV) would have more individual effects on joint risk inference than the parameter in copula. The risk indices (i.e. AND, OR, or Kendall) would not have

significantly influence the individual effects of model parameters. However, for the interactive effects among model parameters, they may exhibit slightly different patterns. Specifically, the parameter in copula would have more significant interactions with parameters in marginal distributions on the failure risk in Kendall than the other two risk indices. Moreover, the individual and interactive effects from model parameters on risk inferences would not influence by the location of gauge stations.


**4.4. Contribution Partition of Uncertainty Sources**

As a result of parameter uncertainties, the predictive failure probabilities exhibit noticeable uncertainties, as shown in Figures 4-6. The three-level factorial analysis based on Equations (11) is able

to provide a primary description and visualization related to the individual and interactive effects of parameter uncertainty on the inferred failure probabilities. However, two critical issue to be answered


are: (i) how much would parameter uncertainties contribute to the variation of the inferred risk values? and (ii) do these contributions change significantly for failure probabilities with different service time scenarios? To address these two issues and get reliable results, a full-subsampling factorial approach

(FSFA) has been proposed, which would be formulated as Equations (16) – (20). Also, similar with the three-level factorial analysis, three quantile levels would be selected at 0.1, 0.5, and 0.9. Based on FSFA, each parameter at its three quantile values (0.1, 0.5, 0.9) would be further subsampled into three scenarios of two quantile values (i.e. (0.1, 0.5), (0.1, 0.9), and (0.5, 0.9)). For this study, we have a total number of 6 parameters with each choose its three quantile values at 0.1, 0.5, and 0.9, which would lead

to a total number of 729 (i.e. $3^6$) two-level factorial designs.

Figure 10 shows the detailed contributions of the model parameters on uncertainty in predictive failure probabilities of AND at the two gauge stations. It can be observed that, even though some discrepancies exist at Zhangjiashan and Xingshan stations, the detailed contributions for each parameter and their

interaction show quite similar features between these two stations. In detail, uncertainty in the shape parameter in GEV has the most significant impact on the failure probability in AND, followed by sdlog in LN, parameter interaction, meanlog in LN, and scale parameter in GEV. Moreover, the uncertainty in the parameter in copula would not lead to significant variation in the resulting failure probability predictions in AND, which would merely make a contribution less than 0.5%. Such conclusions are also

generally consistent to the ANOVA results presented in Figure 7 and Table 4. Furthermore, as the increase in service time, the contributions of each parameter and their interactions would not vary significantly. Some individual contributions from parameter uncertainties would slightly increase while other individual contributions may slightly decrease. However, the effect from parameter interactions would generally increase as the increase of service time. In comparison, the enhancement in design

standard for hydraulic infrastructure would lead to more chance for deceasing in individual effects and, at the same time, increasing in parameter interactions. For instance, as the flood design standard





increases from 200-year to 500-year for a hydraulic facility with 30-year service time near the

Zhangjishan station, the interactive effect of model parameters would increase from 15.14% to 18.09%.

530 ------------------------------------------

Place Figure 10 here

------------------------------------------

In terms of the failure probability in OR, the individual and interactive effects of model parameters on

predictive risk uncertainties show similar pattern with the parameters' effects on the failure probability

in AND. As shown in Figure 11, the shape parameter in the GEV distribution and the sdlog in the LN

distribution are the two major sources for uncertainties in failure probabilities in OR. However,

compared with the failure probability in AND, parameter interaction has a less effect on the resulting

uncertainty of risk inference in OR. As shown in Figures 10 and 11, the effect of parameter interaction

on the risk in AND ranges between 13.96% and 20.05%, while in comparison, the parameters'

interactive effect on the risk in OR varies within [10.25%, 11.57%]. Apparently, it can also be observed

that some external factors such as the design standard and service time of hydraulic infrastructures have

less influence on the parameters' interaction on risk in OR than the risk in AND. However, the first

contributor (i.e. shape parameter in GEV) would have a more contribution on the predictive uncertainty

in the failure probability in OR as the increase in the design standard, while in comparison, this

contributor would have a less contribution on the risk in AND. For instance, as the design return period

of flood (i.e. design standard) increases from 200 to 500 years and the service time of the hydraulic

facility is 30 years, the contribution of the shape parameter in GEV would increase from 47.62% to

50.64% for the failure probability in OR at the Xianyang station, while the parameter's contribution on

the failure probability in AND decreases from 49.26% to 45.77%.



------------------------------------------

Place Figure 11 here

------------------------------------------


For the failure probability in Kendall, the contributions of model parameters and their interaction are
presented in Figure 12. Similar with the failure probabilities in AND and OR, the shape parameter in the
GEV distribution and the sdlog parameter in the LN distribution are the two major contributors, which
can account for nearly 70% or more in the predictive uncertainty of the failure probability in Kendall.

Meanwhile, the scale parameter in GEV, meanlog in LN, and parameters' interaction also have
noticeable effects on the risk in Kendall, ranging from 4.72% (scale parameter in GEV) to 12.64%
(mean log in LN). Conversely, the location parameter in GEV and dependence parameter in copula
merely have quite minor individual effects. However, it is noticeable that, although the dependence
parameter has a minor effect ([0.78%, 1.03%) on the risk in Kendall, such an effect is much higher than

the effect on the risk in AND (less than 0.23%) and the risk in OR (less than 0.06%).

------------------------------------------

Place Figure 12 here

------------------------------------------


Even through the prediction equations for the failure probabilities in AND, OR and Kendall, as
presented in Equations (12) are different, the impacts of parameter uncertainties show quite similar
features, in which the shape parameter in GEV and the sdlog in LN are the two major contributors to the
predictive uncertainties in risk inferences. Nearly 70% and more variability in the uncertainties in risk

inferences can be accounted by the uncertainties in the shape parameter in GEV and sdlog parameter in
LN. Also some external factors such as flood design and facility service time may have different



influence for parameters' effects on different risk indices, such influence is not significant and would not lead to remarkable changes in parameters' contribution to risk inferences. Parameters' interaction has a more effect on risk inference in AND than the other two risk indices (i.e. OR, Kendall), while the

contribution from the dependence parameter, even though not noteworthy, has a more effect on the risk inference in Kendall.

## 5. Discussion

### 5.1. Contribution Partition of Uncertainty Sources through Different Approaches


In current study, the individual and interactive contributions of parameter uncertainties are quantified through the developed FSFA approach, in which each parameter has three levels (i.e. 0.1, 0.5, 0.9 quantiles) to be subsampled. In fact, the parameters' contribution can also be characterized by traditional factorial analysis (FA) approach based on Equations (15) as well as the FSFA approach with

more factor levels (e.g. 4 or 5 levels for each parameter).

Figure 13 shows the comparison of parameter contributions to predictive uncertainty for failure probabilities in AND at Zhangjiashan station for three and four parameter levels scenarios for the design standard of 200-year. The results of Figure 13(b) are obtained through the FSFA approach with each

parameter having four levels to be its quantiles at 0.1, 0.35, 0.6, 0.85. Also, Table 7 presents the parameter contributions to predictive uncertainty in failure probabilities obtained by traditional FA approach for Zhangjiashan stations with the design standard of 200-year and service time of 30-year.

---------------------------------------------

Place Figure 13 and Table 7 here

600 ---------------------------------------------





It can be observed that for different subsampling scenarios, the resulting contributions may be different. However, such a difference would be tolerant since (1) the variations of parameters' contribution are relatively small and mainly happen for the first two contributors, (2) the total contribution of the first two contributors does not change remarkably (around 70% in total), (3) the contributions of other factors especially the parameters' interaction do not vary significantly, and (4) the rank of the contributions from different sources does not change for the two subsampling scenarios. In comparison, as presented in Table 7, the contribution partition of parameter uncertainties obtained through traditional FA shows totally different patterns for different risk inferences. Specifically, traditional FA approach would significantly overestimate parameter interactive effect on risk inference in AND, while at the same time, underestimate the interactive effect on risk inference in OR and Kendall. Consequently, the contribution rank of parameter uncertainties from traditional FA is different from the results obtained through the developed FSFA approach.

As shown in Figure 13, the proposed FSFA approach may lead to slightly different results for different subsampling schemes (four or five levels). However, increase in parameter level would highly increase computational demand. For instance, if each parameter has four levels, FSFA approach would lead to a total number of 46,656 (i.e. $6^6$) two-level factorial designs. Moreover, the subsampling scheme for factors with five levels would lead to a total number of one million (i.e. $6^{10}$) two-level factorial designs. Consequently, the three-level subsampling scheme would generally be recommended and also can generate acceptable results.

## 5.2. Correlation among Parameters' Contributions

The proposed FSFA approach would generally produce a great number of two-level factorial designs. For one specific factor (e.g. GEV_shape), it would have two levels (lower and upper levels) for all



factorial designs. However, the detailed value for the lower or upper level may be different in different

factorial designs. This may finally lead to different contributions for this factor. Figure 14 presents the

variations of parameters' contributions to the prediction of failure probabilities in AND, OR, and

Kendall. We already concluded that the shape parameter in GEV (i.e. GEV_shape) and the sdlog in LN

(i.e. LN_sdlog) distribution would generally have the most significant contributions to predictive

uncertainties in risk inferences. However, as shown in Figure 14, the detailed contributions for these

two parameters would vary remarkably for different level values in different factorial designs. In

comparison, the contributions from other parameters and their interaction have less fluctuation than the

individual contributions of GEV_shape and LN_sdlog. For instance, although the meanlog in LN (i.e.

LN_meanlog), with a average contribution more than 10%, may have some chances to pose a

predominant contribution more than 50%, most of its contribution are positively distributed within [0,

25%]. Also, even though the parameters' interaction has a noteworthy average contribution larger than

10%, all the detailed contributions in different factorial designs are located within [0, 25%].


--------------------------------------------

Place Figure 14 here

--------------------------------------------

It has been observed that the parameters' contribution may vary significantly due to the differences in

factor values in different factorial designs. One potential issue to be addressed is that how those

individual and interactive contributions correlate each other. Figure 15 presents the Pearson's

correlation among individual and interactive contributions of model parameters to different risk

inferences (i.e. failure probabilities in AND, OR, and Kendall). It is noticeable that the parameters in the

LN distribution (i.e. LN_sdlog, LN_meanlog) are generally negatively correlated with the parameters in

the GEV distribution (i.e. GEV_shape, GEV_scale, and GEV_location). Also, for one marginal





distribution (LN or GEV), its parameters are positively correlated. This implies that an increase in the

contribution of one parameter would lead to a contribution increase for parameters within the same

distribution and at the same time result in a contribution decrease for all parameters in the other

distribution. Moreover, if statistically significant, the contribution of the dependent parameter (i.e.

parameter in copula) generally has positive correlation with the contributions from other parameters

except GEV_shape and parameters' interaction. Also, the contribution from parameters' interaction are

generally negatively correlated with the individual contributions from other parameters if such

correlation is statistically significant.

-------------------------------------------

Place Figure 15 here

-------------------------------------------

The proposed FSFA approach can generally characterize how parameter uncertainties would influence

the predictive uncertainties in risk inferences. A large number of two-level factorial designs would be

produced due to different subsampling procedures and then generate different partition results for

parameters' contributions. However, for different risk inferences (i.e. failure probabilities in AND, OR,

and Kendall), these partition results have similar variation features and also show similar correlation

plots.

## 6. Conclusions

Uncertainty quantification is an essential issue for both univariate and multivariate hydrological risk

analysis. A number of research works have been posed to reveal uncertain features in multivariate

hydrological risk inference. However, it is required to know the major sources/contributors for

predictive uncertainties in multivariate risk inferences. In this study, a full-subsampling factorial copula approach (FSFC) has been proposed for uncertainty quantification and partition in multivariate hydrologic risk inference. In FSFC, a copula-based multivariate risk model has been developed and the

bootstrap method is adopted to quantify the probabilistic features for the parameters in both marginal distributions and the dependence model. A full-subsampling factorial analysis (FSFA) approach is finally developed to diminish the effect of the sample size in traditional ANOVA and provided reliable contribution partition for parameter uncertainties in different risk inferences.

This study is the first attempt to characterize parameter uncertainties in a copula-based multivariate hydrological risk model and further reveal their contributions to predictive uncertainties for different risk inferences. As an improvement of ANOVA, the developed FSFA method can mitigate the effect of bias variance estimation in ANOVA and generate reliable results. Moreover, another noteworthy feature for the FSFA approach is that it cannot only characterize the impacts for continuous factors (e.g. model

parameters in this study), but also reveal the impacts of discrete or non-numeric factors. Such a feature can allow the proposed FSFA approach to be employed to further explore the impacts of non-numeric factors (e.g. model structures, sample size) in hydrologic systems analysis.

**Code and data availability:** The flooding data for the studied catchments as well as the associated
code for this study can be gathered upon email request to the corresponding authors

**Author contributions**. YRF, KH, GHH and YPL designed the research. YRF and FW carried out the research, developed the model code and performed the simulations. YRF prepared the manuscript with contributions from all co-authors.


**Competing interests.** The authors declare that they have no conflict of interest.



**Financial support.** This work was jointly funded by the National Key Research and Development Plan
(2016YFC0502800), the National Natural Science Foundation of China (51520105013), and the Natural

Sciences and Engineering Research Council of Canada.

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





**Captions of Tables**

**Table 1.** Flood characteristics for different stations

**Table 2.** Statistical test results for marginal distribution estimation: LN means lognormal distribution, P III means Pearson Type III distribution, and LP III means log-Pearson Type III distribution. K-S test denotes the Kolmogorov–Smirnov test.

**Table 3.** Performance for quantifying the joint distributions between flood peak and volume through

different copulas: CvM is the Cramér von Mises statistic proposed by Genest et al. (2009), with p-value larger than 0.05 indicating satisfactory performance.Table 4. Statistical test results for marginal distribution estimation

**Table 4.** ANOVA table for failure probability in AND: A indicates the shape parameter in GEV, B indicates the scale parameter of GEV, C indicates the location parameter of GEV, D means the meanlog

of LN, E means the sdlog of LN, and F mean the parameters (i.e. theta) in copula Comparison of RMSE and AIC values for joint distributions through different copulas

**Table 5.** ANOVA table for failure probability in OR: A indicates the shape parameter in GEV, B indicates the scale parameter of GEV, C indicates the location parameter of GEV, D means the meanlog of LN, E means the sdlog of LN, and F mean the parameters (i.e. theta) in copula

**Table 6.** ANOVA table for failure probability in Kendall: A indicates the shape parameter in GEV, B indicates the scale parameter of GEV, C indicates the location parameter of GEV, D means the meanlog of LN, E means the sdlog of LN, and F mean the parameters (i.e. theta) in

**Table 7.** Contributions of parameter uncertainties obtained by three level ANOVA to predictive failure probabilities for a design return period of 200-year and service time of 30-year


**Captions of Figures**

**Figure 1:** Framework of the proposed FSFC approach

**Figure 2.** The location of the studied watersheds. Wei River is the largest tributary of Yellow river, with a drainage area of 135,000 km2. The historical flood data from Xianyang and Zhangjiashan stations on

the Wei River are analyzed through the proposed FSFC approach. **Figure 3.** Probabilistic features for parameters in marginal distributions and copula: for both Xianyang and Zhangjiashan stations, the GEV (parameters include shape, scale and location) function would be employed to quantify the distribution of flood peak, while the lognormal distribution (parameters denoted as meanlog and sdlog) is applied for flood volume. The Gumbel and Joe copula (parameter denoted as theta) would be respectively

adopted to model the dependence between flood peak and volume at Zhangjiashan and Xianyang stations.

**Figure 4.** Uncertainty quantification of the joint RP in "AND": the red dash lines indicate the predictive means, the two blue dash lines respectively indicate the 5% and 95% quantiles, and the grey lines indicate the predictions under different parameter samples with the same joint RP of the red and blue



dash lines; The cyan lines denote the predictions under different return periods with the model parameters being their mean values.

**Figure 5.** Uncertainty quantification of the joint RP in "OR": the red dash lines indicate the predictive means, the two blue dash lines respectively indicate the 5% and 95% quantiles, and the grey lines indicate the predictions under different parameter samples with the same joint RP of the red and blue dash lines; The cyan lines denote the predictions under different return periods with the model parameters being their mean values.

**Figure 6.** Uncertainty quantification of the joint RP in "Kendall": the red dash lines indicate the predictive means, the two blue dash lines respectively indicate the 5% and 95% quantiles, and the grey lines indicate the predictions under different parameter samples with the same joint RP of the red and blue dash lines; The cyan lines denote the predictions under different return periods with the model parameters being their mean values

**Figure 7.** Main effects plot and full interactions plot matrix for parameters on the failure probability in AND at the two gauge stations

**Figure 8.** Main effects plot and full interactions plot matrix for parameters on the failure probability in OR at the two gauge stations

**Figure 9.** Main effects plot and full interactions plot matrix for parameters on the failure probability in Kendall at the two gauge stations

**Figure 10.** Contributions of parameter uncertainties to predictive failure probabilities in AND under different design standards (i.e. return periods (RP)) and different service periods

**Figure 11.** Contributions of parameter uncertainties to predictive failure probabilities in OR under different design standards (i.e. return periods (RP)) and different service periods

**Figure 12.** Contributions of parameter uncertainties to predictive failure probabilities in Kendall under different design standards (i.e. return periods (RP)) and different service periods

**Figure 13.** Comparison of parameter contributions to predictive uncertainty for failure probabilities under different levels of subsampling for Zhangjiashan station: three (i.e. 0.1, 0.5, 0.9) and four (i.e. 0.1, 0.35, 0.6, 0.85) level quantiles are adopted for subsampling and the design return period is 200 years.

**Figure 14.** Variation of parameters' contributions for different risk inferences at the Zhangjiashan Station for a design standard of 200-year and a service time of 30-year

**Figure 15.** Correlation for parameters' contributions on risk inferences at Zhangjiashan station for a design standard of 200-year and a service time of 30-year: The cross sign indicates the correlation is statistically insignificant





Table 1. Flood characteristics for different stations

| Station name | period | | flood variable | |
|---|---|---|---|---|
| | | | Peak (m³/s) | Volume (m³/(s day)) |
| | | Minimum | 139 | 317 |
| Xianyang | 1960-2006 | Median | 1350 | 2491 |
| | | Maximum | 12380 | 17802 |
| | | Minimum | 217 | 303.7 |
| Zhangjiashan | 1958-2012 | Median | 775 | 1365.3 |
| | | Maximum | 3730 | 7576.1 |

875



Table 2. Statistical test results for marginal distribution estimation: LN means lognormal distribution, P III means Pearson Type III distribution, and LP III means log-Pearson Type III distribution. K-S test denotes the Kolmogorov–Smirnov test.

| Station name | Flooding variables | Marginal distribution | K-S test $T$ | K-S test P-value | RMSE | AIC |
|---|---|---|---|---|---|---|
| Zhangjiashan | Peak | Gamma | 0.0745 | 0.5471 | 0.0378 | -323.5512 |
| | | **GEV** | **0.0724** | **0.9151** | **0.0275** | **-389.3956** |
| | | LN | 0.0805 | 0.8403 | 0.0283 | -388.3297 |
| | | P III | 0.0893 | 0.7386 | 0.0395 | -349.3274 |
| | | LP III | 0.0795 | 0.8508 | 0.0324 | -371.3614 |
| | Volume | Gamma | 0.1460 | 0.1735 | 0.0596 | -306.2925 |
| | | GEV | 0.1017 | 0.5839 | 0.0369 | -357.0852 |
| | | **LN** | **0.0904** | **0.7250** | **0.0361** | **-361.3353** |
| | | P III | 0.1589 | 0.1112 | 0.0737 | -280.8701 |
| | | LP III | 0.0967 | 0.6468 | 0.0367 | -357.5476 |
| Xianyang | Peak | Gamma | 0.1159 | 0.5533 | 0.0372 | -305.4087 |
| | | **GEV** | **0.0875** | **0.8645** | **0.0305** | **-321.9202** |
| | | LN | 0.1051 | 0.6763 | 0.0436 | -290.5248 |
| | | P III | 0.1202 | 0.5051 | 0.0416 | -292.8448 |
| | | LP III | 0.1321 | 0.3848 | 0.0617 | -255.8931 |
| | Volume | Gamma | 0.1146 | 0.5305 | 0.0450 | -287.4880 |
| | | GEV | 0.0540 | 0.9980 | 0.0195 | -364.3058 |
| | | **LN** | **0.0670** | **0.9749** | **0.0192** | **-367.3885** |
| | | P III | 0.1005 | 0.6913 | 0.0377 | -302.0492 |
| | | LP III | 0.0722 | 0.9522 | 0.0313 | -319.6540 |



Table 3. Performance for quantifying the joint distributions between flood peak and volume through different copulas: CvM is the Cramér von Mises statistic proposed by Genest et al. (2009), with p-value larger than 0.05 indicating satisfactory performance.

|  |  | RMSE | AIC | CvM | p-value |
|---|---|---|---|---|---|
| Zhangjiashan | Gaussian | 0.0669 | -295.5144 | 7.9302 | 0.7770 |
|  | Student t | 0.0669 | -293.5237 | 8.5203 | 0.5976 |
|  | Clayton | 0.0843 | -270.0616 | 9.4615 | 0.3290 |
|  | **Gumbel** | **0.0637** | **-300.8577** | **7.9342** | **0.7580** |
|  | Frank | 0.0690 | -292.0723 | 9.0704 | 0.4480 |
|  | Joe | 0.0606 | -306.3185 | 11.0321 | 0.0290 |
| Xinshan | Gaussian | 0.0513 | -277.1704 | 8.4731 | 0.2400 |
|  | Student t | 0.0510 | -275.6834 | 8.2295 | 0.2885 |
|  | Clayton | 0.0618 | -259.7391 | 8.2051 | 0.3240 |
|  | Gumbel | 0.0477 | -283.9933 | 7.1344 | 0.6700 |
|  | Frank | 0.0562 | -268.6861 | 8.2725 | 0.2940 |
|  | **Joe** | **0.0446** | **-290.2631** | **6.9905** | **0.6540** |





Table 4. ANOVA table for failure probability in AND: A indicates the shape parameter in GEV, B indicates the scale parameter of GEV, C indicates the location parameter of GEV, D means the meanlog of LN, E means the sdlog of LN, and F mean the parameters (i.e. theta) in copula

| Parameter | Zhangjiashan | | | | | Xianyang | | | | |
|---|---|---|---|---|---|---|---|---|---|---|
| | SS | DF | MS | F-Value | P-value | SS | DF | MS | F-Value | P-value |
| A | 0.37 | 2 | 0.18 | 7512.32 | < 0.0001 | 0.59 | 2 | 0.30 | 5079.77 | < 0.0001 |
| B | 0.018 | 2 | 8.905E-003 | 362.65 | < 0.0001 | 0.013 | 2 | 6.527E-003 | 111.71 | < 0.0001 |
| C | 8.313E-005 | 2 | 4.156E-005 | 1.69 | 0.1849 | 8.642E-005 | 2 | 4.321E-005 | 0.74 | 0.4777 |
| D | 0.059 | 2 | 0.029 | 1195.61 | < 0.0001 | 0.082 | 2 | 0.041 | 701.54 | < 0.0001 |
| E | 0.18 | 2 | 0.092 | 3766.70 | < 0.0001 | 0.31 | 2 | 0.16 | 2656.85 | < 0.0001 |
| F | 9.379E-004 | 2 | 4.690E-004 | 19.10 | < 0.0001 | 7.813E-004 | 2 | 3.907E-004 | 6.69 | 0.0013 |
| AB | 2.874E-003 | 4 | 7.186E-004 | 29.26 | < 0.0001 | 8.730E-003 | 4 | 2.183E-003 | 37.35 | < 0.0001 |
| AC | 1.179E-005 | 4 | 2.948E-006 | 0.12 | 0.9753 | 5.434E-005 | 4 | 1.359E-005 | 0.23 | 0.9201 |
| AD | 0.047 | 4 | 0.012 | 473.52 | < 0.0001 | 0.079 | 4 | 0.020 | 338.27 | < 0.0001 |
| AE | 0.14 | 4 | 0.036 | 1448.10 | < 0.0001 | 0.28 | 4 | 0.070 | 1193.40 | < 0.0001 |
| AF | 4.311E-004 | 4 | 1.078E-004 | 4.39 | 0.0017 | 4.687E-004 | 4 | 1.172E-004 | 2.01 | 0.0921 |
| BC | 2.905E-007 | 4 | 7.263E-008 | 2.958E-003 | 1.0000 | 2.235E-008 | 4 | 5.588E-009 | 9.564E-005 | 1.0000 |
| BD | 2.422E-003 | 4 | 6.055E-004 | 24.66 | < 0.0001 | 2.465E-003 | 4 | 6.162E-004 | 10.55 | < 0.0001 |
| BE | 6.956E-003 | 4 | 1.739E-003 | 70.81 | < 0.0001 | 8.669E-003 | 4 | 2.167E-003 | 37.09 | < 0.0001 |
| BF | 8.325E-006 | 4 | 2.081E-006 | 0.085 | 0.9871 | 2.355E-006 | 4 | 5.888E-007 | 0.010 | 0.9998 |
| CD | 1.143E-005 | 4 | 2.859E-006 | 0.12 | 0.9767 | 1.652E-005 | 4 | 4.131E-006 | 0.071 | 0.9909 |
| CE | 3.235E-005 | 4 | 8.088E-006 | 0.33 | 0.8583 | 5.669E-005 | 4 | 1.417E-005 | 0.24 | 0.9142 |
| CF | 3.820E-008 | 4 | 9.551E-009 | 3.889E-004 | 1.0000 | 1.559E-008 | 4 | 3.897E-009 | 6.670E-005 | 1.0000 |
| DE | 1.792E-003 | 4 | 4.481E-004 | 18.25 | < 0.0001 | 6.919E-003 | 4 | 1.730E-003 | 29.60 | < 0.0001 |
| DF | 9.625E-005 | 4 | 2.406E-005 | 0.98 | 0.4178 | 1.288E-004 | 4 | 3.221E-005 | 0.55 | 0.6982 |
| EF | 3.238E-004 | 4 | 8.095E-005 | 3.30 | 0.0109 | 4.540E-004 | 4 | 1.135E-004 | 1.94 | 0.1017 |
| Error | 0.016 | 656 | 2.456E-005 | | | 0.038 | 656 | 5.843E-005 | | |
| Total SS | 0.85 | 728 | | | | 1.42 | 728 | | | |





Table 5. ANOVA table for failure probability in OR: A indicates the shape parameter in GEV, B indicates the scale parameter of GEV, C indicates the location parameter of GEV, D means the meanlog of LN, E means the sdlog of LN, and F mean the parameters (i.e. theta) in copula

| Parameter | Zhangjiashan | | | | | Xianyang | | | | |
|---|---|---|---|---|---|---|---|---|---|---|
| | SS | DF | MS | F-Value | P-value | SS | DF | MS | F-Value | P-value |
| A | 2.04 | 2 | 1.02 | 39285.40 | < 0.0001 | 3.71 | 2 | 1.85 | 30534.64 | < 0.0001 |
| B | 0.20 | 2 | 0.098 | 3784.17 | < 0.0001 | 0.26 | 2 | 0.13 | 2165.79 | < 0.0001 |
| C | 9.466E-004 | 2 | 4.733E-004 | 18.22 | < 0.0001 | 1.811E-003 | 2 | 9.054E-004 | 14.91 | < 0.0001 |
| D | 0.24 | 2 | 0.12 | 4679.22 | < 0.0001 | 0.30 | 2 | 0.15 | 2498.09 | < 0.0001 |
| E | 0.60 | 2 | 0.30 | 11626.79 | < 0.0001 | 0.87 | 2 | 0.43 | 7132.20 | < 0.0001 |
| F | 7.833E-004 | 2 | 3.916E-004 | 15.08 | < 0.0001 | 6.382E-004 | 2 | 3.191E-004 | 5.26 | 0.0054 |
| AB | 0.17 | 4 | 0.043 | 1666.34 | < 0.0001 | 0.27 | 4 | 0.069 | 1128.63 | < 0.0001 |
| AC | 8.076E-004 | 4 | 2.019E-004 | 7.77 | < 0.0001 | 1.830E-003 | 4 | 4.575E-004 | 7.54 | < 0.0001 |
| AD | 0.048 | 4 | 0.012 | 465.73 | < 0.0001 | 0.081 | 4 | 0.020 | 335.01 | < 0.0001 |
| AE | 0.15 | 4 | 0.037 | 1418.17 | < 0.0001 | 0.29 | 4 | 0.071 | 1175.81 | < 0.0001 |
| AF | 3.442E-004 | 4 | 8.604E-005 | 3.31 | 0.0106 | 3.658E-004 | 4 | 9.144E-005 | 1.51 | 0.1986 |
| BC | 8.013E-005 | 4 | 2.003E-005 | 0.77 | 0.5442 | 1.226E-004 | 4 | 3.064E-005 | 0.50 | 0.7323 |
| BD | 2.528E-003 | 4 | 6.319E-004 | 24.33 | < 0.0001 | 2.534E-003 | 4 | 6.334E-004 | 10.43 | < 0.0001 |
| BE | 7.212E-003 | 4 | 1.803E-003 | 69.40 | < 0.0001 | 8.837E-003 | 4 | 2.209E-003 | 36.39 | < 0.0001 |
| BF | 5.294E-006 | 4 | 1.323E-006 | 0.051 | 0.9951 | 1.077E-006 | 4 | 2.693E-007 | 4.436E-003 | 1.0000 |
| CD | 1.192E-005 | 4 | 2.981E-006 | 0.11 | 0.9773 | 1.697E-005 | 4 | 4.242E-006 | 0.070 | 0.9911 |
| CE | 3.353E-005 | 4 | 8.382E-006 | 0.32 | 0.8628 | 5.780E-005 | 4 | 1.445E-005 | 0.24 | 0.9169 |
| CF | 2.416E-008 | 4 | 6.040E-009 | 2.325E-004 | 1.0000 | 7.119E-009 | 4 | 1.780E-009 | 2.932E-005 | 1.0000 |
| DE | 0.11 | 4 | 0.028 | 1069.79 | < 0.0001 | 0.17 | 4 | 0.042 | 691.60 | < 0.0001 |
| DF | 7.482E-005 | 4 | 1.870E-005 | 0.72 | 0.5784 | 9.919E-005 | 4 | 2.480E-005 | 0.41 | 0.8026 |
| EF | 2.568E-004 | 4 | 6.420E-005 | 2.47 | 0.0435 | 3.550E-004 | 4 | 8.876E-005 | 1.46 | 0.2121 |
| Error | 0.017 | 656 | 2.598E-005 | | | 0.040 | 656 | 6.071E-005 | | |
| Total SS | 3.60 | 728 | | | | 6.01 | 728 | | | |



Table 6. ANOVA table for failure probability in Kendall: A indicates the shape parameter in GEV, B indicates the scale parameter of GEV, C indicates the location parameter of GEV, D means the meanlog of LN, E means the sdlog of LN, and F mean the parameters (i.e. theta) in copula

| Parameter | Zhangjiashan | | | | | Xianyang | | | | |
|---|---|---|---|---|---|---|---|---|---|---|
| | SS | DF | MS | F-Value | P-value | SS | DF | MS | F-Value | P-value |
| A | 0.97 | 2 | 0.48 | 33813.15 | < 0.0001 | 2.08 | 2 | 1.04 | 27047.85 | < 0.0001 |
| B | 0.096 | 2 | 0.048 | 3349.45 | < 0.0001 | 0.15 | 2 | 0.076 | 1983.83 | < 0.0001 |
| C | 4.627E-004 | 2 | 2.313E-004 | 16.19 | < 0.0001 | 1.055E-003 | 2 | 5.274E-004 | 13.72 | < 0.0001 |
| D | 0.11 | 2 | 0.057 | 3987.58 | < 0.0001 | 0.17 | 2 | 0.084 | 2181.53 | < 0.0001 |
| E | 0.28 | 2 | 0.14 | 9809.61 | < 0.0001 | 0.47 | 2 | 0.24 | 6153.63 | < 0.0001 |
| F | 0.013 | 2 | 6.451E-003 | 451.42 | < 0.0001 | 0.025 | 2 | 0.013 | 331.58 | < 0.0001 |
| AB | 0.087 | 4 | 0.022 | 1525.14 | < 0.0001 | 0.16 | 4 | 0.041 | 1066.18 | < 0.0001 |
| AC | 4.090E-004 | 4 | 1.022E-004 | 7.15 | < 0.0001 | 1.101E-003 | 4 | 2.754E-004 | 7.16 | < 0.0001 |
| AD | 0.022 | 4 | 5.448E-003 | 381.22 | < 0.0001 | 0.044 | 4 | 0.011 | 286.00 | < 0.0001 |
| AE | 0.066 | 4 | 0.017 | 1156.06 | < 0.0001 | 0.15 | 4 | 0.038 | 995.79 | < 0.0001 |
| AF | 2.163E-003 | 4 | 5.407E-004 | 37.84 | < 0.0001 | 5.986E-003 | 4 | 1.496E-003 | 38.93 | < 0.0001 |
| BC | 4.233E-005 | 4 | 1.058E-005 | 0.74 | 0.5645 | 7.800E-005 | 4 | 1.950E-005 | 0.51 | 0.7304 |
| BD | 1.147E-003 | 4 | 2.866E-004 | 20.06 | < 0.0001 | 1.377E-003 | 4 | 3.444E-004 | 8.96 | < 0.0001 |
| BE | 3.254E-003 | 4 | 8.135E-004 | 56.93 | < 0.0001 | 4.755E-003 | 4 | 1.189E-003 | 30.93 | < 0.0001 |
| BF | 2.479E-004 | 4 | 6.198E-005 | 4.34 | 0.0018 | 4.939E-004 | 4 | 1.235E-004 | 3.21 | 0.0126 |
| CD | 5.408E-006 | 4 | 1.352E-006 | 0.095 | 0.9842 | 9.226E-006 | 4 | 2.307E-006 | 0.060 | 0.9933 |
| CE | 1.513E-005 | 4 | 3.782E-006 | 0.26 | 0.9007 | 3.112E-005 | 4 | 7.781E-006 | 0.20 | 0.9370 |
| CF | 1.190E-006 | 4 | 2.974E-007 | 0.021 | 0.9992 | 3.369E-006 | 4 | 8.423E-007 | 0.022 | 0.9991 |
| DE | 0.054 | 4 | 0.014 | 950.06 | < 0.0001 | 0.096 | 4 | 0.024 | 623.48 | < 0.0001 |
| DF | 1.870E-004 | 4 | 4.676E-005 | 3.27 | 0.0114 | 3.437E-004 | 4 | 8.592E-005 | 2.24 | 0.0638 |
| EF | 4.113E-004 | 4 | 1.028E-004 | 7.20 | < 0.0001 | 9.130E-004 | 4 | 2.282E-004 | 5.94 | 0.0001 |
| Error | 9.374E-003 | 656 | 1.429E-005 | | | 0.025 | 656 | 3.844E-005 | | |
| Total SS | 1.72 | 728 | | | | 3.40 | 728 | | | |



Table 7. Contributions of parameter uncertainties obtained by three level ANOVA to predictive failure probabilities for a design return period of 200-year and service time of 30-year

| Factor | FPand | FPor | FPkendall |
|---|---|---|---|
| A | 43.53% | 56.67% | 56.40% |
| B | 2.12% | 5.56% | 5.58% |
| C | 0.01% | 0.03% | 0.03% |
| D | 6.94% | 6.67% | 6.40% |
| E | 21.18% | 16.67% | 16.28% |
| F | 0.11% | 0.02% | 0.76% |
| Interaction | 26.12% | 4.72% | 5.06% |



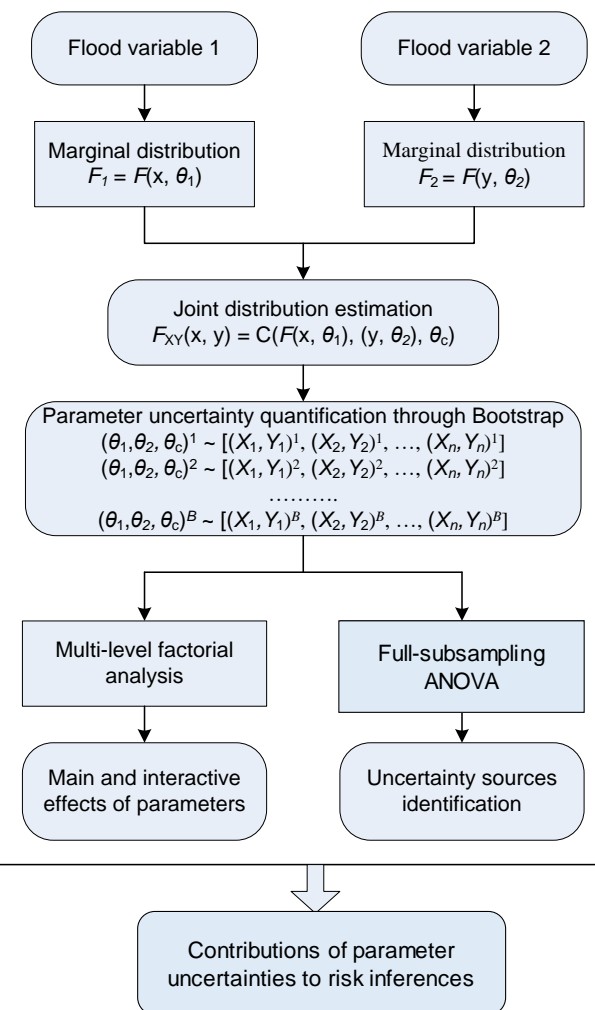

Figure 1. Framework of the proposed FSFC approach.

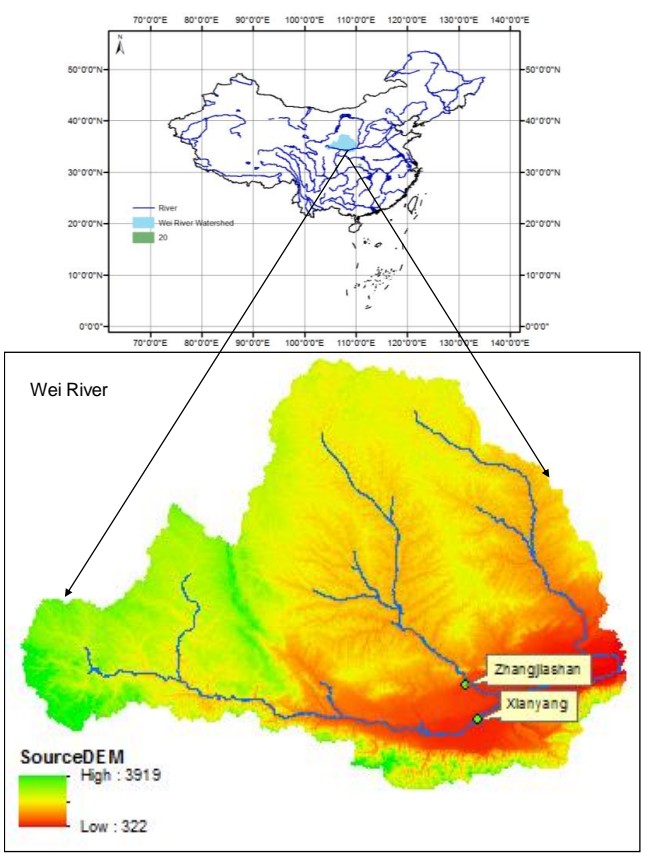

Figure 2. The location of the studied watersheds. Wei River is the largest tributary of Yellow river, with a drainage area of $135,000$ km$^2$. The historical flood data from Xianyang and Zhangjiashan stations on the Wei River are analyzed through the proposed FSFC approach.



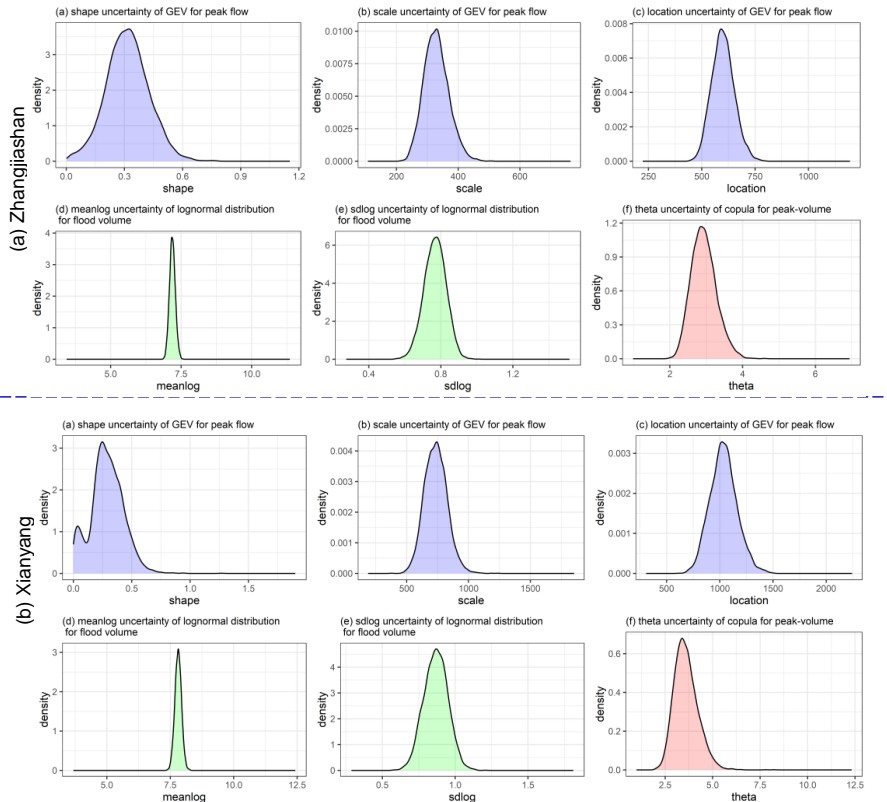

Figure 3. Probabilistic features for parameters in marginal distributions and copula: for both Xianyang and Zhangjiashan stations, the GEV (parameters include shape, scale and location) function would be employed to quantify the distribution of flood peak, while the lognormal distribution (parameters denoted as meanlog and sdlog) is applied for flood volume. The Gumbel and Joe copula (parameter denoted as theta) would be respectively adopted to model the dependence between flood peak and volume at Zhangjiashan and Xianyang stations.

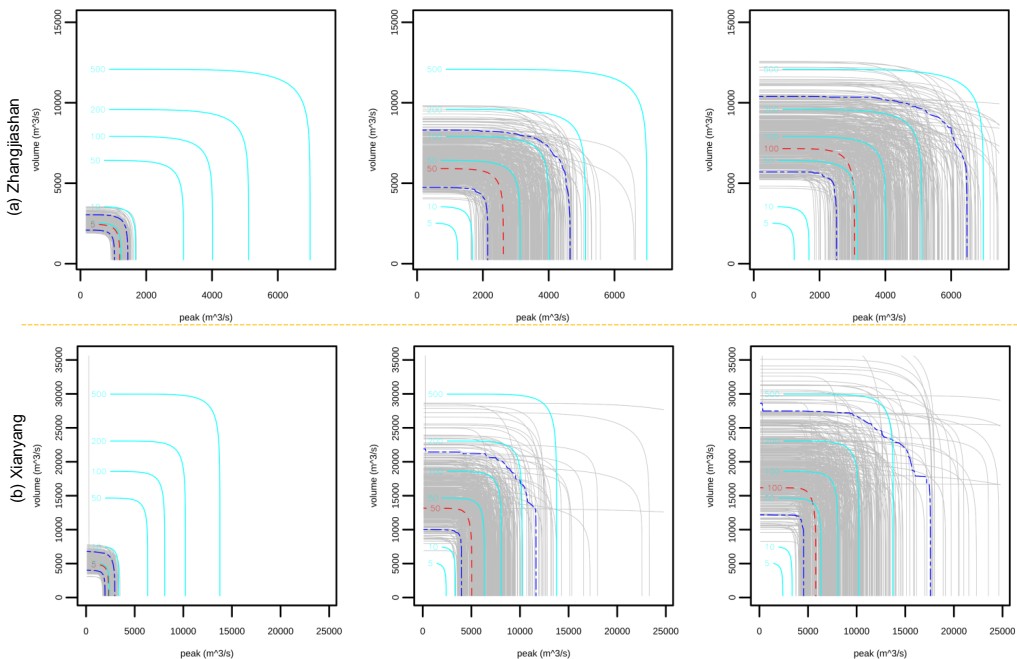

Figure 4. Uncertainty quantification of the joint RP in "AND": the red dash lines indicate the predictive means, the two blue dash lines respectively indicate the 5% and 95% quantiles, and the grey lines indicate the predictions under different parameter samples with the same joint RP of the red and blue dash lines; The cyan lines denote the predictions under different return periods with the model parameters being their mean values.

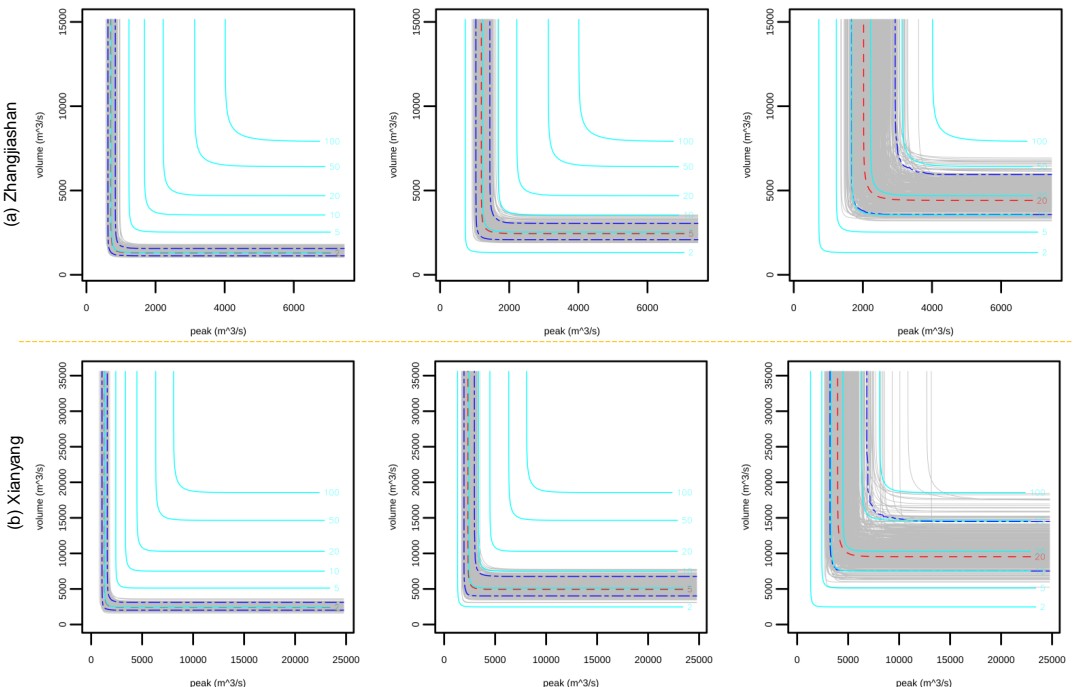

Figure 5. Uncertainty quantification of the joint RP in "OR": the red dash lines indicate the predictive means, the two blue dash lines respectively indicate the 5% and 95% quantiles, and the grey lines indicate the predictions under different parameter samples with the same joint RP of the red and blue dash lines; The cyan lines denote the predictions under different return periods with the model parameters being their mean values.



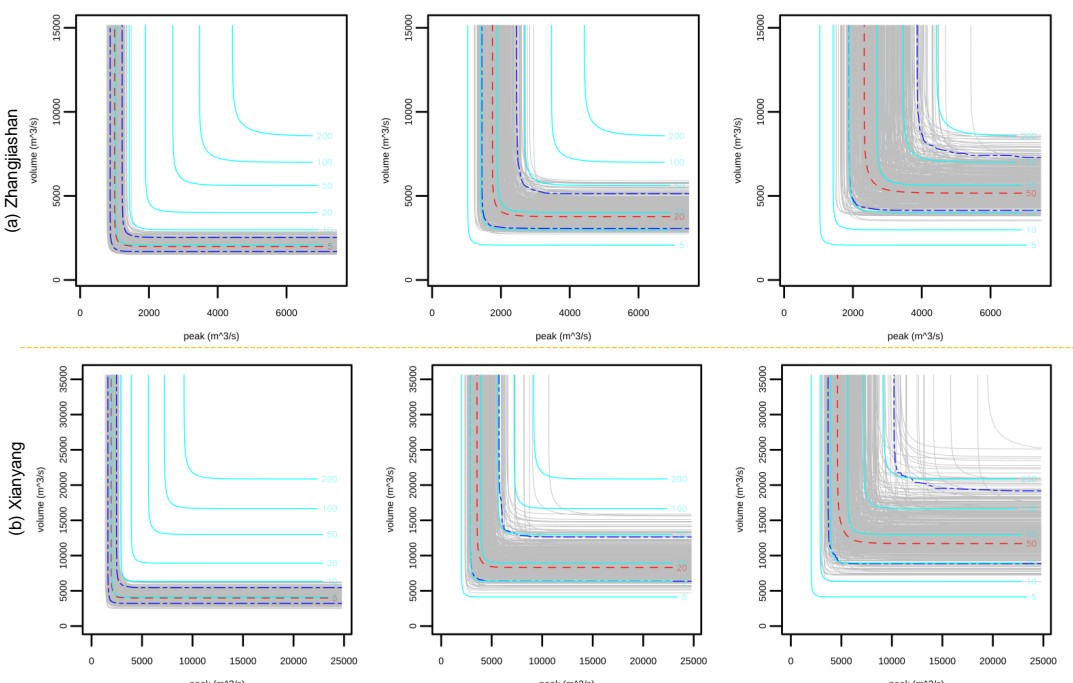

Figure 6. Uncertainty quantification of the joint RP in "Kendall": the red dash lines indicate the predictive means, the two blue dash lines respectively indicate the 5% and 95% quantiles, and the grey lines indicate the predictions under different parameter samples with the same joint RP of the red and blue dash lines; The cyan lines denote the predictions under different return periods with the model parameters being their mean values.

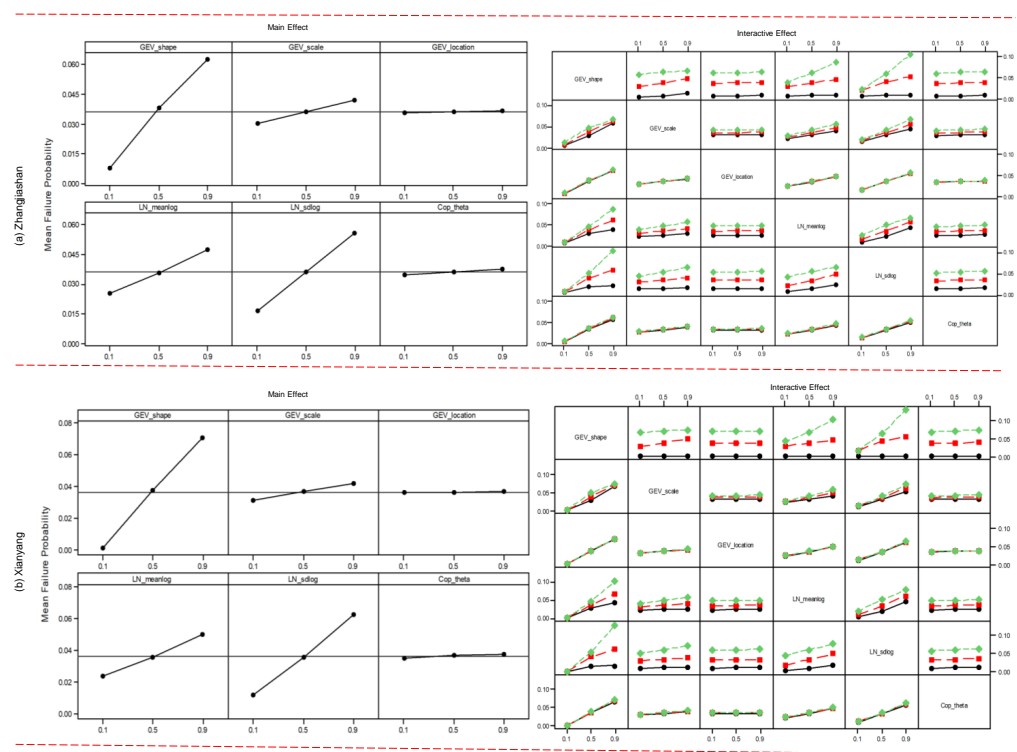

Figure 7. Main effects plot and full interactions plot matrix for parameters on the failure probability in AND at the two gauge stations.



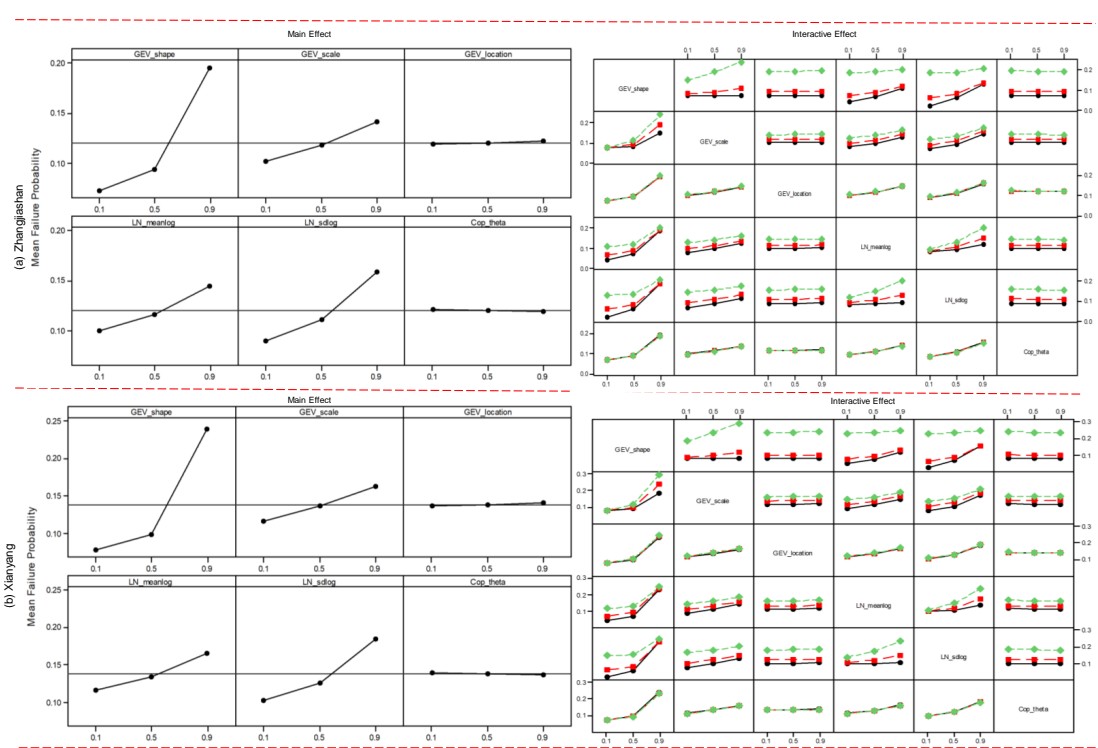

Figure 8. Main effects plot and full interactions plot matrix for parameters on the failure probability in OR at the two gauge stations



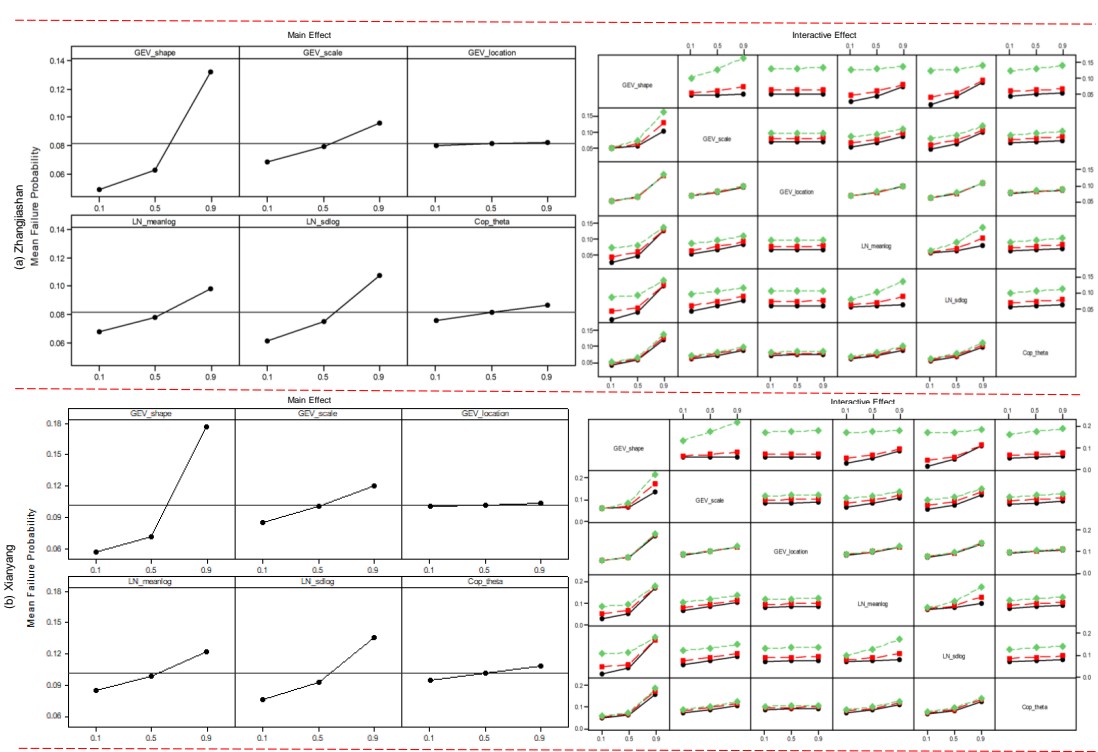

Figure 9. Main effects plot and full interactions plot matrix for parameters on the failure probability in Kendall at the two gauge stations







Figure 10. Contributions of parameter uncertainties to predictive failure probabilities in AND under different design standards (i.e. return periods (RP)) and different service periods





| | | (a) Zhangjiashan | | | (b) Xianyang | | |
|---|---|---|---|---|---|---|---|
| **RP: 200** | GEV_shape | 45.60% | 45.64% | 45.52% | 47.62% | 47.53% | 47.22% |
| | GEV_scale | 6.94% | 6.57% | 6.23% | 5.23% | 4.77% | 4.37% |
| | GEV_location | 0.07% | 0.07% | 0.06% | 0.07% | 0.06% | 0.06% |
| | LN_meanlog | 12.60% | 12.55% | 12.50% | 10.96% | 10.92% | 10.86% |
| | LN_sdlog | 24.48% | 24.75% | 24.98% | 24.51% | 24.97% | 25.36% |
| | Cop_theta | 0.06% | 0.06% | 0.06% | 0.03% | 0.03% | 0.03% |
| | Par_interaction | 10.25% | 10.37% | 10.65% | 11.57% | 11.73% | 12.11% |
| **RP: 300** | GEV_shape | 46.95% | 47.05% | 47.05% | 49.05% | 49.07% | 48.96% |
| | GEV_scale | 6.47% | 6.16% | 5.87% | 5.04% | 4.64% | 4.28% |
| | GEV_location | 0.05% | 0.05% | 0.04% | 0.05% | 0.05% | 0.04% |
| | LN_meanlog | 11.65% | 11.61% | 11.58% | 10.11% | 10.07% | 10.04% |
| | LN_sdlog | 24.37% | 24.63% | 24.86% | 24.19% | 24.61% | 25.00% |
| | Cop_theta | 0.05% | 0.04% | 0.04% | 0.02% | 0.02% | 0.02% |
| | Par_interaction | 10.47% | 10.47% | 10.55% | 11.54% | 11.54% | 11.66% |
| **RP: 500** | GEV_shape | 48.45% | 48.56% | 48.63% | 50.64% | 50.73% | 50.74% |
| | GEV_scale | 6.01% | 5.77% | 5.54% | 4.85% | 4.53% | 4.24% |
| | GEV_location | 0.03% | 0.03% | 0.03% | 0.04% | 0.03% | 0.03% |
| | LN_meanlog | 10.65% | 10.62% | 10.60% | 9.22% | 9.20% | 9.17% |
| | LN_sdlog | 24.14% | 24.35% | 24.56% | 23.73% | 24.07% | 24.40% |
| | Cop_theta | 0.03% | 0.03% | 0.03% | 0.02% | 0.02% | 0.02% |
| | Par_interaction | 10.69% | 10.63% | 10.61% | 11.51% | 11.43% | 11.41% |
| | | 30 year | 50 year | 70 year | 30 year | 50 year | 70 year |

Legend: 1%, 10%, 20%, 30%, 40%, 50%

Figure 11. Contributions of parameter uncertainties to predictive failure probabilities in OR under different design standards (i.e. return periods (RP)) and different service periods







Figure 12. Contributions of parameter uncertainties to predictive failure probabilities in Kendall under different design standards (i.e. return periods (RP)) and different service periods





Figure 13. Comparison of parameter contributions to predictive uncertainty for failure probabilities under different levels of subsampling for Zhangjiashan station: three (i.e. 0.1, 0.5, 0.9) and four (i.e. 0.1, 0.35, 0.6, 0.85) level quantiles are adopted for subsampling and the design return period is 200 years.



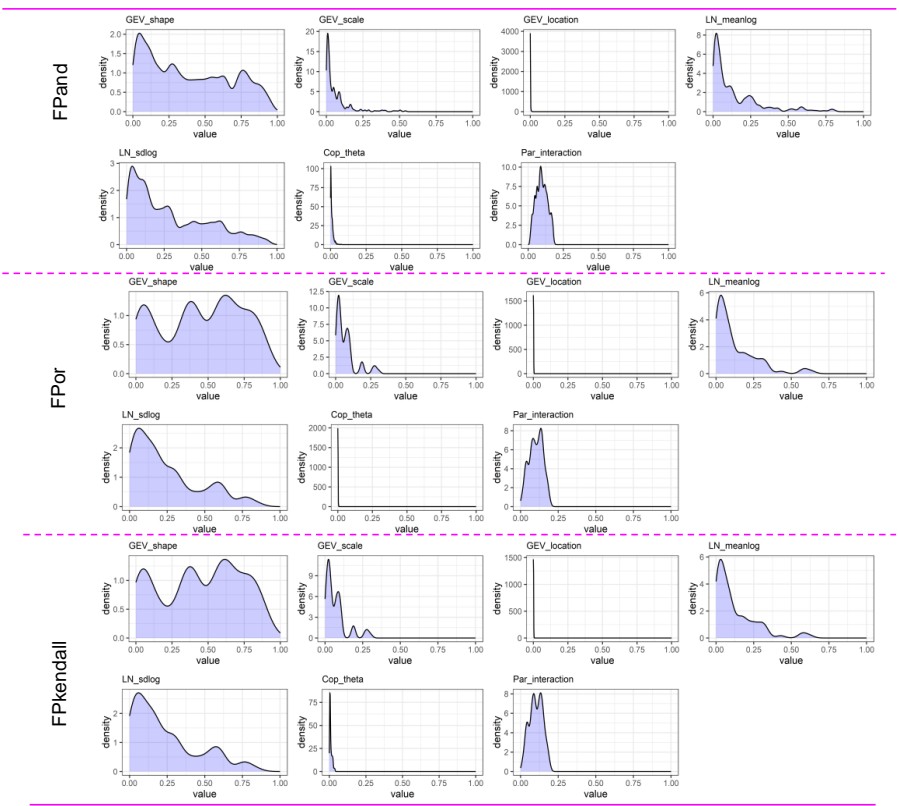

Figure 14. Variation of parameters' contributions for different risk inferences at the Zhangjiashan
Station for a design standard of 200-year and a service time of 30-year





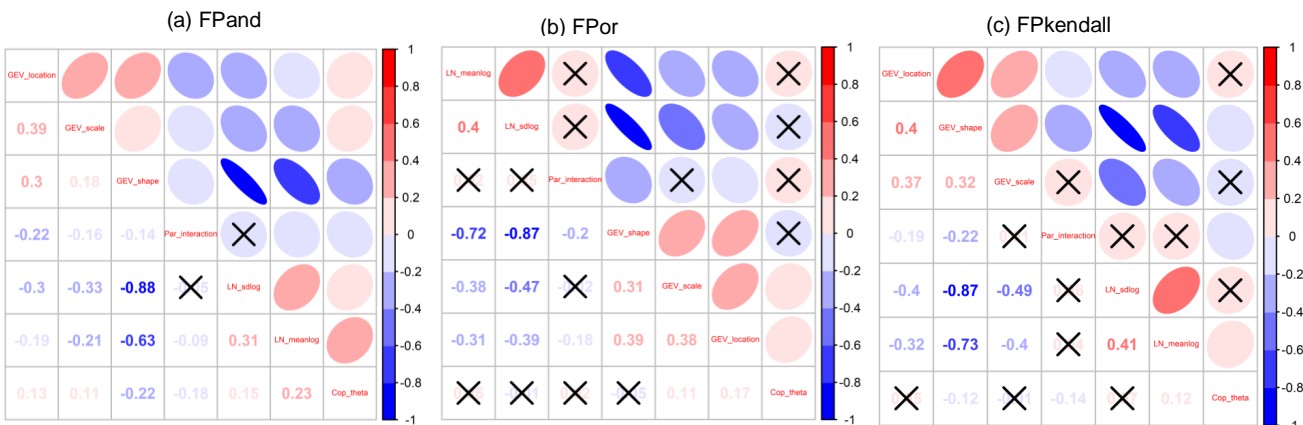

Figure 15. Correlation for parameters' contributions on risk inferences at Zhangjiashan station for a design standard of 200-year and a service time of 30-year: The cross sign indicates the correlation is statistically insignificant.