# Peer review of "An Uncertainty Partition Approach for Inferring Interactive Hydrologic Risks"

_Hydrology and Earth System Sciences, 2019_

## Referee Comment (RC1) · Anonymous Referee #1 · 6 Nov 2019

This is a very good paper with some very interesting results. The authors have conducted the onerous task of evaluating the uncertainty in parameters (that may or may not be interacting) that describe probability distributions typically used in flood prediction. They have selected appropriate methods for characterizing those uncertainties (FSFA approach), describing the contribution to uncertainty and laid out all the relevant equations in as succinct a way as possible. Everything is clearly outlined and the results are clearly stated. The paper shines in particular in the way the results are presented visually. A wide range of measures (parameters and their paired interaction) are evaluated creating a large number of results. The visual presentation of these results is excellent. This is no easy task but they managed to convey the outcomes graphically in an easily accessible way that makes this kind of research accessible overall (where

generally it is not due to the forest of indices and equations). On that note, there are minor technical errors mostly in language that I will not list here; only one error in the equation I could find (line 331, the dot normally indicating placement of index "l" should not be there) and a question on whether all those significant digits are actually warranted in Tables 2-6 and in Figures 10-13. Figure 1 could be expanded I would argue to explicitly indicate copulas, etc and provide a few more details on the framework. The work as a whole is nice and succinct – although, greater discussion in the discussion section could be given in view of the importance of this work. Thus, I ask the authors just two questions to consider here explaining in their discussion section: 1. The two watersheds selected are not very different but you find discrepancies between which copulas perform best on which stations (lines 409 to 412) for predicting flood peak and volume, and different copulas are chosen to characterize uncertainty in the risks for each station (line 414-416). The authors are using data driven methods that have no explicit consideration for causal mechanisms (as with most data driven methods) but surely the differences in copulas selected are caused by physical differences in the watersheds. Can the authors please explain these discrepancies in terms of physical watershed characteristics (or perhaps make the case for why the differences cannot be ascribed to physical differences)? 2. Please detail what part of the analyses is watershed specific and thus, what analysis should each user conduct each time and for every station they wish to understand prediction uncertainty and parameter interaction with the outcomes of their analyses; or conversely, what can they simply adopt from the Tables and Figures for their watersheds?

---

## Referee Comment (RC2) · Geoff Pegram (Referee) · 9 Mar 2020

Fan_hess-2019-434

Review

The paper is a thorough and deep stochastic study of a pair of high river flows and their extrapolation to design periods, deriving useful estimates of their accuracy and reliability, exploring a wide range of modelling possibilities.

I have been careful to suggest some cosmetic improvements to the text, although I confess to suffering some 'symbol shock' in section 2.4! I cannot fault the mathematics and am impressed by the depth of detail that the authors have gone to, in order to be absolutely sure that their deductions are sound. Although only two stations in China

were used as exemplars, the methodology will be invaluable in regional assessments of catchments' high flow characteristics.

I am uploading my detailed comments in the marked up version of the manuscript for the authors to address and leave it at that.

Geoff Pegram

09 March 2020

Please also note the supplement to this comment:
https://www.hydrol-earth-syst-sci-discuss.net/hess-2019-434/hess-2019-434-RC2-supplement.pdf

[Figure]

**Supplement:**

[revised manuscript text omitted]

this is the width of HESS margins

please expand this map about 3 times - on an A4 page the lettering is illegible

[Figure]

Figure 2. The location of the studied watersheds. Wei River is the largest tributary of Yellow river, with a drainage area of 135,000 km$^2$. The historical flood data from Xianyang and Zhangjiashan stations on the Wei River are analyzed through the proposed FSFC approach.

[Figure]

[Figure]

this is the width of
HESS margins

please expand this
figure - on an A4 page
the lettering is very
small and hard to read

[Figure]

Figure 3. Probabilistic features for parameters in marginal distributions and copula: for both Xianyang and Zhangjiashan stations, the GEV (parameters include shape, scale and location) function would be employed to quantify the distribution of flood peak, while the lognormal distribution (parameters denoted as meanlog and sdlog) is applied for flood volume. The Gumbel and Joe copula (parameter denoted as theta) would be respectively adopted to model the dependence between flood peak and volume at Zhangjiashan and Xianyang stations.

[Figure]

please expand this figure - on an A4 page the lettering is very small

[Figure]

Figure 4. Uncertainty quantification of the joint RP in "AND": the red dash lines indicate the predictive means, the two blue dash lines respectively indicate the 5% and 95% quantiles, and the grey lines indicate the predictions under different parameter samples with the same joint RP of the red and blue dash lines; The cyan lines denote the predictions under different return periods with the model parameters being their mean values.

[Figure]

[Figure]

[Figure]

Figure 5. Uncertainty quantification of the joint RP in "OR": the red dash lines indicate the predictive means, the two blue dash lines respectively indicate the 5% and 95% quantiles, and the grey lines indicate the predictions under different parameter samples with the same joint RP of the red and blue dash lines; The cyan lines denote the predictions under different return periods with the model parameters being their mean values.

[Figure]

[Figure]

please expand this figure - on an A4 page the lettering is very small

[Figure]

Figure 6. Uncertainty quantification of the joint RP in "Kendall": the red dash lines indicate the predictive means, the two blue dash lines respectively indicate the 5% and 95% quantiles, and the grey lines indicate the predictions under different parameter samples with the same joint RP of the red and blue dash lines; The cyan lines denote the predictions under different return periods with the model parameters being their mean values.

[Figure]

[Figure]

please expand this figure - on an A4 page the lettering is very small and the font should be increased in size - it seems this request applies to all figures, but not tables.

[Figure]

Figure 7. Main effects plot and full interactions plot matrix for parameters on the failure probability in AND at the two gauge stations.

in figures 7 through 9, please add some text in the caption to explain what the figures are showing the reader. In the images of 'interactive effect', why is the labelling put on the diagonal? are the labels describing vertical columns or horizontal rows? what are the green, red and black plots telling us? This layout poses an unnecessary puzzle for the reader.

[revised manuscript text omitted]

---

## Author Comment (AC1) · 17 Mar 2020

Manuscript ID: hess-2019-434

RESPONSES TO REVIEWER #1's COMMENTS

We are grateful to Reviewer #1 for his/her insightful review. The provided comments have contributed substantially to improving the paper. According to them, we have made significant efforts to revise the manuscript, with the details explained as follows:

Point #1
*COMMENT:*
*only one error in the equation I could find (line 331, the dot normally indicating placement of index "l" should not be there)*

RESPONSE: We are thankful for the reviewer's carefulness, and have corrected this part as follows:
Figure 1 show the locations of these three gauging stations based on the daily stream flow data

Point #2
*COMMENT:*
*a question on whether all those significant digits are actually warranted in Tables 2-6 and in Figures 10-13.*
.
RESPONSE: We appreciate the reviewer's comment. The digits in Tables 2-3 and Figures 10-13 are rounded in Excel, and the digits in Tables 4-6 are generated by the Design-Experts. All those significant digits can be warranted.

Point #3
*COMMENT:*
*Figure 1 could be expanded I would argue to explicitly indicate copulas, etc and provide a few more details on the framework.*

RESPONSE: We are grateful for the reviewer's suggestion. Firstly, we rename the full-subsampling factorial copula method to iterative factorial copula, which is more concise. Also, the full-subsampling factorial analysis is renamed as iterative factorial analysis. Also, we have provided more details for Figure 1 as follows:

Figure 1 illustrates the framework of the proposed IFC approach. The framework consists of four modules: (i) selection of marginal distributions, (ii) identification of copulas, (iii) parameter uncertainty quantification, (iv) parameter interaction and sensitivity analysis. In IFC, modules (i) and (ii) are proposed to construct the most appropriate copula-based hydrologic risk model. In detail, a number of distributions, such as Gamma, generalized extreme value (GEV), lognormal (LN), Pearson type III (P III), and log-Pearson type III (LP III) distributions, are usually employed to describe the probabilistic features of individual random variables (e.g. flood peak and volume). Also, in order to quantify the dependence structures of correlated random variables, many copula functions have been proposed, such as Gaussian copula, Student t copula, Archimedean copula family (e.g. Clayton, Gumbel, Frank and Joe copulas). In

the current study, the indices of root mean square errors (RMSE) and Akaike information criterion (AIC) will be employed to identify the most appropriate model for hydrologic risk inference. Module (iii) quantifies parameter uncertainties in marginal distributions and copulas. Modules (iv) would be the core part of our study to identify the main sources of uncertainties in multivariate risk inference by the proposed iterative factorial analysis (IFA) approach.

Point #4
*COMMENT:*
*The two watersheds selected are not very different but you find discrepancies between which copulas perform best on which stations (lines 409 to 412) for predicting flood peak and volume, and different copulas are chosen to characterize uncertainty in the risks for each station (line 414-416). The authors are using data driven methods that have no explicit consideration for causal mechanisms (as with most data driven methods) but surely the differences in copulas selected are caused by physical differences in the watersheds. Can the authors please explain these discrepancies in terms of physical watershed characteristics (or perhaps make the case for why the differences cannot be ascribed to physical differences)?*

RESPONSE: We are thankful for the reviewer's suggestion. We have added discussion for this issue in Section 5.1 as follows:

**5.1. Differences for the Hydrologic Risk Models at Different Stations**

Different copula functions are applied for different stations, which are chosen based on the indices of RMSE and AIC. However, the selection of copula models at different stations may also be related with some key characteristics of the drainage areas for those stations. The Gumbel copula will be applied for the Zhangjiashan station. It can reflect strong correlation at high values. However, the Joe copula, which is adopted for the Xianyang station, can reflect stronger right tail positive dependence. Both the Xianyang and Zhangjiashan stations have similar drainage areas. The Xianyang station controls a drainage area of 46,480 km$^2$ (Xu et al., 2016), while the Zhangjiashan station has a drainage area of 45,412 km$^2$ (Sun et al., 2019). Nevertheless, the major reason that lead to different copula functions for these two stations may be due to the elevation features for those two drainage areas. The drainage area of Zhangjiashan station is located in the central part of Loess Plateau of China and thus the major part of this drainage area is a mountainous region. In comparison, even thought a large part of the drainage area of Xianyang station is also located in the mountainous region, the Xianyang station also controls a significant part of the Guanzhong Plain, as indicated in the red part of Figure 2. Consequently, the flood hydrograph at Zhangjiashan station may be sharp while the flood hydrograph at Xianyang station is relatively flat and show a stronger right tail dependence among flood peak and volume. In fact, the value of Kendall's tau between peak and volume for the top ten floods at Zhangjiashan station is 0.33 while such a value of Kendall's tau at Xianyang station is 0.6. These facts may explain the Gumbel copula is applicable for Zhangjiashan station while the Joe copula is applied for Xianyang station.

Point #4

*COMMENT:*
*Please detail what part of the analyses is watershed specific and thus, what analysis should each user conduct each time and for every station they wish to understand prediction uncertainty and parameter interaction with the outcomes of their analyses; or conversely, what can they simply adopt from the Tables and Figures for their watersheds*

RESPONSE: We are grateful for the reviewer's suggestion. This study aims to propose a reliable uncertainty partition method for multivariate risk inference. Based on this method, the decision maker can track the major sources for the uncertainties in the risk inferences. The proposed method can be applied for different watersheds. We have highlight the usefulness of our study in conclusions as follows:

> The proposed method has been applied for flood risk inferences at two gauge stations in Wei River basin. The results indicate that uncertainties in the parameters of the copula-based model would lead to noticeable uncertainties in the resulting risk inferences, especially for the joint flood risk in AND. noticeable uncertainties exist in the predictive joint RP of AND even for a small flood event. However, the results from IFA suggested that those uncertainties in risk inferences may mainly be attributed to the uncertainties in shape parameter in GEV distribution and the parameter of sdlog in LN for both the two stations. In comparison, the parameter uncertainty in the copula function would not pose an obvious effect on the resulting uncertainty in risk inferences. Such results indicate that, at least that the Wei River basin, the decision makers need to well estimate the values or quantify the uncertainties for the shape parameter in GEV distribution and sdlog in the LN distribution, in order to obtain reliable risk inferences. For other catchments, the proposed IFC method can be adopted to reveal the major sources for uncertainties in risk inferences and then provide potential pathways to get reliable risk inferences.

---

## Author Comment (AC2) · 17 Mar 2020

Manuscript ID: hess-2019-434

RESPONSES TO REVIEWER #2's COMMENTS

We are grateful to Reviewer #2 for his/her insightful review. The provided comments have contributed substantially to improving the paper. According to them, we have made significant efforts to revise the manuscript, with the details explained as follows:

Point #1
*COMMENT:*
*detailed comments in the marked up version*

RESPONSE: We are really thankful for the reviewer's carefulness and comments. We have revised all the errors in our manuscript carefully which have been highlighted in the revision

Point #2
*COMMENT:*
*There is something missing here. Please introduce your 'OR' and 'AND' cases to assist the reader.   Also, what are the names of $T^{OR}$ and $T^{AND}$? what do they describe?   What is $R^d$?*

RESPONSE: We appreciate the reviewer's comment. We have provided more details about the joint return period in OR and AND as follows:

(i) "OR" case $T^{OR}$:

$$T^{OR} = \{(x_1, x_2, ..., x_d) \in R^d : x_1 > x_1^* \vee x_2 > x_2^* \vee ... \vee x_d > x_d^*\}$$
$$= \frac{\mu}{1 - C(F_1(x_1 \mid \gamma_1), ..., F_d(x_d \mid \gamma_d) \mid \theta)} \tag{4}$$

where $R^d$ is a d-dimensional real space; $\mu$ denotes the average time between two adjacent events under consideration. The joint RP in OR (denoted as $T^{OR}$) indicates the occurrence probability of the extreme event with one of its variables $x_i$'s, $i = 1, 2, ..., d$, exceeding the corresponding threshold $x_i^*$

(ii) "AND" case $T^{AND}$:

$$T^{AND} = \{(x_1, x_2, ..., x_d) \in R^d : x_1 > x_1^* \wedge x_2 > x_2^* \wedge ... \wedge x_d > x_d^*\}$$
$$= \frac{\mu}{\hat{C}(\overline{F}_1(x_1 \mid \gamma_1), \overline{F}_2(x_2 \mid \gamma_2), ..., \overline{F}_1(x_d \mid \gamma_d) \mid \theta)} \tag{5}$$

where $\hat{C}$ is the multivariate survival function of the $X_i$'s proposed by Salvadori et al. (2013;

2016), and $\overline{F}_i(x_i \mid \gamma_i) = P(X > x_i) = 1 - F_i(x_i \mid \gamma_i)$. Following Salvadori et al. (2013; 2016), and

the Inclusion-Exclusion principle proposed by Joe (2014), the multivariate survival function $\overset{\wedge}{C}$ can be obtained by:

$$\overset{\wedge}{C}(\mathbf{u}) = \overline{C}(1-\mathbf{u}) \tag{6}$$

and

$$\overline{C}(\mathbf{u}) = 1 - \sum_{i=1}^{d} u_i + \sum_{S \in P} (-1)^{\#(S)} C_S(u_i : i \in S) \tag{7}$$

where $\#(S)$ denotes the cardinality of $S$. The joint RP in AND (denoted as $T^{AND}$) of the extreme event indicates the occurrence probability with all of its variables $x_i$'s, $i = 1, 2, \ldots, d$, exceeding the corresponding thresholds $x_i^*$'s

Point #3
*COMMENT:*
*In order to be accurate, Figs 10 to 12 are tables and should be labelled and referred to as such*

RESPONSE: We are grateful for the reviewer's comment. Figures 10 – 12 show the contributions of model parameters to uncertainties in risk inferences under different design standards and service time periods. These results seem to be presented in tables. However, Figures 10-12 are in fact heat maps generated by Excel, in which different parameter contributions are highlighted by different colors. Consequently, we labelled them as figures rather than tables.